



# Precipitation Susceptibility of Marine Stratocumulus with Variable Above and Below-Cloud Aerosol Concentrations over the Southeast Atlantic

Siddhant Gupta[1,2], Greg M. McFarquhar[1,2], Joseph R. O'Brien[3], Michael R. Poellot[3], David J. Delene[3], Rose M. Miller[4], and Jennifer D. Small Griswold[5]

[1]Cooperative Institute for Mesoscale Meteorological Studies, University of Oklahoma, Norman, OK, USA
[2]School of Meteorology, University of Oklahoma, Norman, OK, USA
[3]Department of Atmospheric Sciences, University of North Dakota, Grand Forks, ND, USA
[4]Department of Atmospheric Sciences, University of Illinois at Urbana-Champaign, Urbana, IL,
USA
[5]Department of Meteorology, University of Hawai'i at Manoa, Honolulu, HI, USA

*Correspondence to:* Siddhant Gupta (sid@ou.edu)

**Abstract.** Aerosol-cloud-precipitation interactions (ACIs) provide the greatest source of

uncertainties in predicting changes in Earth's energy budget due to poor representation of

marine stratocumulus and the associated ACIs in climate models. Using in situ data from 329

cloud profiles across 24 research flights from the NASA ObseRvations of Aerosols above CLouds

and their intEractionS (ORACLES) field campaign in September 2016, August 2017, and October

2018, it is shown that contact between above-cloud biomass-burning aerosols and marine

stratocumulus over the southeast Atlantic Ocean was associated with precipitation suppression

and a decrease in the precipitation susceptibility ($S_o$) to aerosols. The 173 "contact" profiles with

aerosol concentration ($N_a$) greater than 500 cm$^{-3}$ within 100 m above cloud tops had 50 % lower

precipitation rate ($R_p$) and 20 % lower $S_o$, on average, compared to 156 "separated" profiles with

$N_a$ less than 500 cm$^{-3}$ up to at least 100 m above cloud tops.

Contact and separated profiles had statistically significant differences in droplet

concentration ($N_c$) and effective radius ($R_e$) (95 % confidence intervals from a two-sample t-test

are reported). Contact profiles had 84 to 90 cm$^{-3}$ higher $N_c$ and 1.4 to 1.6 μm lower $R_e$ compared

to separated profiles. In clean boundary layers (below-cloud $N_a$ less than 350 cm$^{-3}$), contact profiles had 25 to 31 cm$^{-3}$ higher $N_c$ and 0.2 to 0.5 μm lower $R_e$. In polluted boundary layers

(below-cloud $N_a$ exceeding 350 cm$^{-3}$), contact profiles had 98 to 108 cm$^{-3}$ higher $N_c$ and 1.6 to 1.8 μm lower $R_e$. On the other hand, contact and separated profiles had statistically insignificant differences between the average liquid water path, cloud thickness, and meteorological parameters like surface temperature, lower tropospheric stability, and estimated inversion strength. These results suggest the changes in cloud properties were driven by ACIs rather than

meteorological effects, and the existing relationships between $R_p$ and $N_c$ must be adjusted to account for the role of ACIs.

## 1 Introduction

Clouds drive the global hydrological cycle with an annual average precipitation rate of 3 mm day$^{-1}$ over the oceans (Behrangi et al., 2014). Marine stratocumulus (MSC) is the most

common cloud type with an annual coverage of 22 % over the ocean surface (Eastman et al., 2011). These low-level, boundary layer clouds typically exist over subtropical oceans in regions with large-scale subsidence such as the southeast Atlantic Ocean (Klein and Hartmann, 1993). MSC have higher reflectivity (albedo) than the ocean surface which results in a strong, negative shortwave cloud radiative forcing (CRF) with a weak and positive longwave CRF (Oreopoulos and

Rossow, 2011).

Low-cloud cover in the subsidence regions is negatively correlated with sea surface temperature (SST) (Eastman et al., 2011; Wood and Hartmann, 2006). CRF is thus sensitive to changes in SST but there is a large spread in model estimates of CRF sensitivity (Bony and



Dufresne, 2005). This provides uncertainty in the model estimates of Earth's energy budget in

future climate scenarios (Trenberth and Fasullo, 2009). Uncertainty in parameterization of

boundary layer aerosol, cloud, and precipitation processes contributes to model uncertainties

(Ahlgrimm and Forbes, 2014; Stephens et al., 2010).

MSC CRF is regulated by cloud processes that depend on cloud microphysical properties,

like droplet concentration ($N_c$), effective radius ($R_e$), and liquid water content (LWC), and

macrophysical properties, like cloud thickness ($H$) and liquid water path (LWP). These cloud

properties can depend on the concentration, composition, and size distributions of aerosols

which act as cloud condensation nuclei. Under conditions of constant LWC, increases in aerosol

concentration ($N_a$) can increase $N_c$ and decrease $R_e$, strengthening the shortwave CRF (Twomey,

1974, 1977). A decrease in droplet sizes in polluted clouds can inhibit droplet growth from

collision-coalescence and suppress precipitation intensity, resulting in lower precipitation rate

($R_p$), higher LWP, and increased cloud lifetime (Albrecht, 1989). In combination, these aerosol-

cloud-precipitation interactions (ACIs) and the resulting cloud adjustments lead to an effective

radiative forcing termed $ERF_{aci}$ (Boucher et al., 2013).

Satellite retrievals of $R_e$ and cloud optical thickness ($\tau$) can be used to estimate $N_c$ and

LWP using the adiabatic assumption (Boers et al., 2006; Wood and Hartmann, 2006; Bennartz,

2007). LWC increases linearly with height in adiabatic clouds and $\tau$ is parameterized as a function

of $N_c$ and LWP ($\tau \propto N_c^{1/3}$ LWP$^{5/6}$) (Brenguier et al., 2000). Since $\tau$ has greater sensitivity to LWP

compared to $N_c$, assuming constant LWP can lead to underestimation of the cloud albedo

susceptibility to aerosol perturbations (Platnick and Twomey, 1994).



LWP can have a positive or negative response to increasing $N_c$ due to aerosols (Toll et al., 2019). The LWP response is regulated by environmental conditions (e.g., lower tropospheric stability (LTS), boundary layer depth ($H_{BL}$), and relative humidity), cloud particle sizes (e.g., represented by $R_e$), $R_p$, and by $N_c$ and LWP themselves (Chen et al., 2014; Gryspeerdt et al., 2019; Toll et al., 2019; Possner et al., 2020). Accurate estimation of the LWP response to aerosol

perturbations is important for regional and global estimates of $ERF_{aci}$ (Douglas and L'Ecuyer, 2019; 2020).

      Droplet evaporation associated with cloud-top entrainment and precipitation are the two major sinks of LWP in MSC. Smaller droplets associated with higher $N_c$ or $N_a$ evaporate more readily which leads to greater cloud-top evaporative cooling and a negative LWP response (Hill

et al., 2008). The LWP response to the evaporation-entrainment feedback (Xue and Feingold, 2006; Small et al., 2009) also depends on above-cloud humidity (Ackerman et al., 2004). Precipitation susceptibility ($S_o$) to aerosol-induced changes in cloud properties is defined as the change in $R_p$ due to aerosol-induced changes in $N_c$ and is a function of LWP or $H$ (Feingold and Seibert, 2009).

The magnitude of $S_o$ depends on precipitation formation processes like collision-coalescence which are parameterized using mass transfer rates, such as the autoconversion rate ($S_{AUTO}$) and the accretion rate ($S_{ACC}$) (Morrison and Gettelman, 2008; Geoffroy et al., 2010). Autoconversion describes the process of collisions between cloud droplets that coalesce to form drizzle drops which initiate precipitation. Accretion refers to collisions between cloud droplets

and drizzle drops which lead to larger drizzle drops and greater precipitation intensity. The



variability in $S_o$ as a function of LWP or $H$ depends on the cloud type and the ratio of $S_{ACC}$ versus $S_{AUTO}$ (Wood et al., 2009; Jiang et al., 2010; Sorooshian et al., 2010).

Recent studies of ACIs have focused on the southeast Atlantic Ocean because of the unique meteorological conditions present in the region (Zuidema et al., 2016; Redemann et al.,

2021). Biomass-burning aerosols from southern Africa are transported over an extensive MSC deck that exists off the coast of Namibia and Angola (Adebiyi and Zuidema, 2016; Devasthale and Thomas, 2011). The aerosol layer is comprised of shortwave-absorbing aerosols (500 nm single-scattering albedo of about 0.83) and high above-cloud aerosol optical depth (up to 0.42) (Pistone et al., 2019; LeBlanc et al., 2020). The sign of the forcing due to shortwave absorption by the

aerosol layer depends on the location of aerosols in the vertical column and the albedo of the underlying clouds (Cochrane et al., 2019).

Satellite retrievals suggest warming aloft due to a positive forcing decreases dry air entrainment into clouds, increases LWP and cloud albedo, and decreases the shortwave CRF (Wilcox, 2010). The net radiative forcing due to the aerosol and cloud layers thus depends on

aerosol-induced changes in $N_c$, $R_e$, and LWP and the resulting changes in $\tau$. Sinks of $N_c$ and LWP like precipitation and entrainment mixing lead to uncertainties in satellite retrievals of $N_c$ which pose the biggest challenge in the use of satellite retrievals to study the aerosol impact on $N_c$ (Quaas et al., 2020). This motivates observational studies of ACIs that examine $N_c$ and LWP under different aerosol and meteorological conditions.

In situ observations of cloud and aerosol properties were collected over the southeast Atlantic during the NASA ObseRvations of Aerosols above CLouds and their intEractionS



(ORACLES) field campaign during three Intensive Observation Periods (IOPs) in September 2016, August 2017, and October 2018 (Redemann et al., 2021). The above-cloud aerosol plume was associated with elevated water vapor content (Pistone at al., 2021) which influenced cloud-top humidity and dynamics following the mechanisms discussed by Ackerman et al. (2004).

During the 2016 IOP, variable vertical displacement (0 to 2000 m) was observed between above-cloud aerosols and the MSC (Gupta et al., 2021; hereafter G21). Instances of contact and separation between the aerosol and cloud layers were associated with differences in the above- and below-cloud $N_a$, water vapor mixing ratio ($w_v$), and cloud-top entrainment processes. These differences led to changes in $N_c$, $R_e$, and LWC, and their vertical profiles (G21). In this study, the response of the MSC to above- and below-cloud aerosols is further examined using data from all three ORACLES IOPs, and precipitation formation and $S_o$ are evaluated as a function of $H$.

The paper is organized as follows. In Section 2, the ORACLES observations are discussed along with the data quality assurance procedures (additional details are in a supplement). In Section 3, the calculation of cloud properties is described. In Section 4, the influence of aerosols on $N_c$, $R_e$, and LWC is examined by comparing the parameters for MSC in contact or separated from the above-cloud aerosol layer. In Section 5, the changes in precipitation formation due to aerosol-induced microphysical changes are examined. In Section 6, $N_c$, $R_p$, and $S_o$ are examined as a function of $H$ and the above- and below-cloud $N_a$. In Section 7, the meteorological conditions are examined using reanalysis data. In Section 8, the conclusions are summarized with directions for future work.



## 2 Observations

The ORACLES IOPs were based at Walvis Bay, Namibia (23° S, 14.6° E) in September 2016,

and at São Tomé and Príncipe (0.3° N, 6.7° E) in August 2017 and October 2018. The data analyzed

in this study were collected during the three IOPs (Table 1 and Figure 1): six P-3 research flights

(PRFs) from 6 to 25 September 2016 with cloud sampling conducted between 1° W to 12° E and

9° S to 20° S; seven PRFs from 12 to 28 August 2017 with cloud sampling conducted between 8°

W to 6° E and 2° S to 15° S; and 11 PRFs from 27 September to 23 October 2018 with cloud

sampling conducted between 3° W to 9° E and 1° N to 15° S. These PRFs were selected because

in situ cloud sampling was conducted during at least three vertical profiles through the cloud

layer (Table 1).

Three PRFs from the 2016 IOP had overlapping tracks when the P-3B aircraft flew north-

west from 23° S, 13.5° E toward 10° S, 0° E, and returned along the same track (Figure 1). The

2017 and 2018 IOPs had 10 PRFs with overlapping flight tracks when the aircraft flew south from

0° N, 5° E toward 15° S, 5° E, and returned along the same track. PRFs with overlapping tracks

acquired statistics for model evaluation (Doherty et al., 2021) while the other PRFs targeted

specific locations based on meteorological conditions (Redemann et al., 2021).

During ORACLES, the NASA P-3B aircraft was equipped with in situ probes. The data

analyzed in this study were collected using Cloud Droplet Probes (CDPs), a Cloud and Aerosol

Spectrometer (CAS) on the Cloud, Aerosol and Precipitation Spectrometer, a Phase Doppler

Interferometer (PDI), a Two-Dimensional Stereo Probe (2D-S), a High Volume Precipitation

Sampler (HVPS-3), a King hot-wire, and a Passive Cavity Aerosol Spectrometer Probe (PCASP). A



single CDP was used during the 2016 IOP (hereafter CDP-A), a second CDP (hereafter CDP-B) was

added for the 2017 and 2018 IOPs, and CDP-A was replaced by a different CDP (hereafter CDP-C)

for the 2018 IOP.

The CAS, CDP, King hot-wire, and PCASP data were processed at the University of North

Dakota using the Airborne Data Processing and Analysis processing package (Delene, 2011). The

*PDI* data were processed at the University of Hawai'i. The 2D-S and HVPS-3 data were processed

using the University of Illinois/Oklahoma Optical Array Probe Processing Software (McFarquhar

et al., 2018). The data processing procedures followed to reject artifacts were summarized by

G21. Comparisons between the cloud probe data sets are described in the supplement.

The King hot-wire was used to sample LWC (hereafter King LWC). The PCASP was used to

sample the accumulation-mode aerosols sized from 0.1 to 3.0 μm. The PCASP *N(D)* was used to

determine the out-of-cloud $N_a$. The CAS, CDP, PDI, 2D-S, and HVPS-3 collectively sampled the

number distribution function *N(D)* for particles with diameter *D* from 0.5 to 19200 μm. The size

distribution covering the complete droplet size range was determined by merging the *N(D)* for 3

< *D* < 50 μm with the *N(D)* for 50 < *D* < 1050 μm from the *2D-S* and the *N(D)* for 1050 < *D* < 19200

μm from the HVPS-3. The HVPS-3 sampled droplets with *D* > 1050 μm for a single 1 Hz data

sample across the PRFs analyzed in this study.

During each PRF, at least two independent measurements of *N(D)* were made for 3 < *D* <

50 μm using the CAS, the PDI or a CDP (Table 1). The differences between the $N_c$ and LWC derived

from the CAS, PDI and CDP *N(D)* were quantified. The LWC estimates from the CAS, PDI, and CDP

were compared with the adiabatic LWC (LWC$_{ad}$) which represents the theoretical maximum for





175 LWC (Brenguier et al., 2000). The $N(D)$ for droplets with $D < 50$ µm was determined using the

probe which consistently had the LWC with better agreement with the $LWC_{ad}$ during each IOP.

The differences between in-cloud data sets from different instruments were determined

using a two-sample t-test. The 95 % confidence intervals (CIs) between parameter means were

reported if the differences were statistically significant. During the 2017 IOP, the CAS and the

180 CDP-B sampled droplets with $D < 50$ µm. The CDP-B LWC was higher than the CAS LWC (95 % CIs:

0.11 to 0.12 g m$^{-3}$ higher), and the average CDP-B LWC (0.18 g m$^{-3}$) had better agreement with

the average $LWC_{ad}$ (0.24 g m$^{-3}$) compared to the average CAS LWC (0.08 g m$^{-3}$). Thus, the CDP-B

$N(D)$ was used to represent the $N(D)$ for droplets with $D < 50$ µm for the 2017 IOP.

Similar results were obtained when the CAS LWC and the CDP-B LWC were compared with

185 the $LWC_{ad}$ for the 2018 IOP. During the 2018 IOP, the CDP-C was mounted at a different location

relative to the aircraft wing compared to the CAS and CDP-B, and the positions of CDP-B and CDP-

C were switched after 10 October 2018. O'Brien et al. (2021, in prep) found the CDP mounting

positions had only a 6 % impact on the calculation of $N_c$ and the average CDP-B LWC and CDP-C

LWC were within 0.02 g m$^{-3}$. To maintain consistency with the 2017 IOP, data from the CDP

190 mounted next to the CAS were used for droplets with $D < 50$ µm for the 2018 IOP (except on 15

October 2018 when the CDP-C had a voltage issue).

During the 2016 IOP, measurements from the CDP-A were unusable for all PRFs due to an

optical misalignment issue. Nevertheless, the CAS and the PDI sampled droplets with $3 < D < 50$

µm. On average, the PDI LWC was higher than the CAS LWC (95 % CIs: 0.20 to 0.21 g m$^{-3}$ higher).

195 Since the PDI LWC was greater than the $LWC_{ad}$ (95 % CIs: 0.04 to 0.06 g m$^{-3}$ higher), it was



hypothesized that the PDI LWC was an overestimate of the actual LWC. Thus, the *CAS N(D)* was used to represent the *N(D)* for droplets with *D* < 50 µm for the 2016 IOP.

The *2D-S* has two channels which concurrently sample the cloud volume. $N_c$ and LWC were derived using data from the horizontal channel ($N_H$ and LWC$_H$) and the vertical channel ($N_V$

and LWC$_V$). $N_H$ and LWC$_H$ were used for the 2016 IOP because $N_V$ and LWC$_V$ were not available due to soot deposition on the inside of the receive-side mirror of the vertical channel. $N_H$ and $N_V$ as well as LWC$_H$ and LWC$_V$ were strongly correlated for the 2017 and 2018 IOPs with Pearson's correlation coefficient $R \geq 0.92$ and the best-fit slope $\geq 0.90$. The high correlation values suggest that little difference would have resulted from using the average of the two 2D-S channels. To

maintain consistency with the 2016 *IOP*, $N_H$ and LWC$_H$ were used for all three IOPs.

## 3 Cloud Properties

The *N(D)* from the merged droplet size distribution was integrated to calculate $N_c$. The 1 Hz data samples with $N_c > 10$ cm$^{-3}$ and King *LWC* > 0.05 g m$^{-3}$ were defined as in-cloud measurements (G21). In situ cloud sampling during ORACLES included flight legs when the P-3B

aircraft ascended or descended through the cloud layer (hereafter cloud profiles). Data from 329 cloud profiles with just under four hours of cloud sampling were examined (Table 1).

For every cloud profile, the cloud top height ($Z_T$) was defined as the highest altitude with $N_c > 10$ cm$^{-3}$ and King LWC > 0.05 g m$^{-3}$ (Table 2). The average $Z_T$ during ORACLES was 1038 ± 270 m, where the uncertainty estimate refers to the standard deviation. Possner et al. (2020) found

that investigations of MSC in boundary layers shallower than 1 km can provide an underestimate of the LWP adjustments associated with ACIs. $Z_T$ was used as a proxy for boundary layer height



and the average $Z_T$ greater than 1 km suggests these measurements represent the complete range of LWP adjustments associated with ACIs.

The cloud base height ($Z_B$) was defined as the lowest altitude with $N_c > 10$ cm$^{-3}$ and King
LWC $> 0.05$ g m$^{-3}$. In decoupled boundary layers, a layer of cumulus can be present below the stratocumulus layer with a gap between the cloud layers (Wood, 2012). Measurements from stratocumulus were used in this study and $Z_B$ for the stratocumulus layer was identified as the altitude above which the King LWC increased without gaps greater than 25 m in the cloud sampling up to $Z_T$.

The difference between $Z_T$ and $Z_B$ was defined as $H$. Due to aerosol-induced changes in entrainment and boundary layer stability, the aerosol impact on $H$ and $Z_T$ can have the strongest influence on LWP adjustments associated with ACIs (Toll et al., 2019). Thus, the influence of ACIs on precipitation formation and $S_o$ was examined as a function of $H$. Data collected during incomplete profiles of the stratocumulus or while sampling open-cell clouds (for example, on 2$^{nd}$
October 2018) were excluded because of difficulties with estimating $H$ for such profiles.

For each 1 Hz in-cloud data sample, the droplet size distribution was used to calculate $R_e$ following Hansen and Travis (1974), where,

$$R_e = \int_3^\infty D^3\, N(D)\, dD \Big/ \int_3^\infty 2\, D^2\, N(D)\, dD\,. \tag{1}$$

LWC was calculated as

$$LWC\,(h) = \pi\, \rho_w / 6 \int_3^\infty D^3\, N(D,h)\, dD\,, \tag{2}$$





where $\rho_w$ is the density of liquid water and $h$ is height in cloud above cloud base. LWC and King LWC were integrated over $h$ from $Z_B$ to $Z_T$ to calculate LWP and King LWP, respectively. $\tau$ was calculated as

$$\beta_{ext}\,(h) = \int_3^\infty Q_{ext}\,\pi/4\,D^2\,N(D,h)\,dD\,,\ \ \tau = \int_{Z_B}^{Z_T}\beta_{ext}\,(h)\,dh\,, \qquad (3)$$

where $\beta_{ext}$ is the cloud extinction and $Q_{ext}$ is the extinction coefficient (approximately 2 for cloud droplets assuming geometric optics apply for visible wavelengths) (Hansen and Travis, 1974). The integrals in Eq. (1) to (3) were converted to discrete sums for $D > 3\ \mu m$ to consider the contributions of cloud drops, and not aerosols.

The total water mixing ratio ($w_t$) in a cloud is the sum of $w_v$ and the liquid water mixing

ratio ($w_l$). At cloud base, $w_v = w_s$, where $w_s$ is the saturation water vapor mixing ratio. $w_l$ and $w_s$ at $Z_B$ were calculated as

$$w_l(Z_B) = LWC(Z_B)/\rho_a\,,\ w_v(Z_B) = w_s = \ 1000\,\epsilon\,e_s\,(T,Z_B)/p(Z_B) - e_s(T,Z_B)\,, \qquad (4)$$

where $\rho_a$ is the density of air, $\varepsilon$ is the ratio of the gas constants of air and water vapor, $p$ is pressure, and $e_s$ is the saturation water vapor pressure which depends on temperature ($T$). $e_s$

and $w_s$ decrease with $h$ because $T$ decreases with $h$ following the moist adiabatic lapse rate. For adiabatic clouds, $w_t$ is constant and the adiabatic $w_l$ increases with height as $w_s$ decreases (the subscript 'ad' is added hereafter to denote adiabatic values). $w_{lad}$ was multiplied by $\rho_a$ to calculate $LWC_{ad}$. According to the adiabatic model (Brenguier et al., 2000), $LWC_{ad}$ and $LWP_{ad}$ are functions of $H$. These relationships help parameterize $\tau_{ad}$ as

$LWC_{ad}(h) = C_w\,h\,,\ LWP_{ad} = 1/2\,C_w\,H^2\,,\ \tau_{ad} \propto (\alpha\,C_w)^{-1/6}\,(kN_c)^{1/3}\,LWP^{5/6}\,, \qquad (5)$



where $C_w$ is condensation rate, $\alpha$ is cloud adiabaticity (LWP divided by $LWP_{ad}$), and $k$ is droplet spectrum width (Brenguier et al., 2000). $C_w$ is a function of the cloud base $p$ and $T$ (Brenguier et al., 2000) and $\alpha$ helps quantify the impact of entrainment mixing or precipitation on cloud water. Assuming constant $C_w$ (1.44 to 2) or $\alpha$ (0.6 to 1) can lead to errors in satellite

retrievals of $N_c$ (Janssen et al., 2011; Merk et al., 2016; Grosvenor et al., 2018) which motivates the need for in situ estimates of $C_w$ and $\alpha$. $C_w$ was calculated using a regression model to fit $LWP_{ad}$ as a function of $H$. $LWP_{ad}$ was a quadratic function of $H$ (Figure 2) with $R \geq 0.93$. The average $C_w$ for the three IOPs was $2.71 \pm 0.30$ g m$^{-3}$ km$^{-1}$ (Table 3). This was greater than $C_w$ for MSC over the northeast Pacific (2.33 g m$^{-3}$ km$^{-1}$) (Braun et al., 2018).

For 304 cloud profiles with $LWP_{ad} > 5$ g m$^{-2}$, the average $\alpha$ was $0.72 \pm 0.31$ ($0.85 \pm 0.41$ if the King hot-wire was used to represent LWC). This was consistent with $\alpha$ for MSC over the northeast Pacific ($0.77 \pm 0.13$) (Braun et al., 2018) and the southeast Pacific (median $\alpha = 0.7$ to 0.8) (Min et al., 2012). The differences between $LWP_{ad}$ and LWP increased with $H$. For example, when the profiles were divided into thin ($H < 175$ m) and thick clouds ($H > 175$ m) based on the

median $H$, thin clouds had higher $\alpha$ ($0.84 \pm 0.34$) than thick clouds ($0.60 \pm 0.23$). The inverse relationship between $\alpha$ and $H$ is consistent with previous MSC observations (Braun et al., 2018).

**4 Aerosol Influence on Cloud Microphysics**

The MSC over the southeast Atlantic were overlaid by biomass-burning aerosols from southern Africa (Adebiyi and Zuidema, 2016; Redemann et al., 2021) with instances of contact

and separation between the MSC cloud tops and the base of the biomass burning aerosol layer (G21). Across the three IOPs, 173 profiles were conducted at locations where an extensive

aerosol plume with $N_a > 500$ cm$^{-3}$ was located within 100 m above $Z_T$ (hereafter, contact profiles) (Table 1). 156 profiles were conducted at locations where the level of $N_a > 500$ cm$^{-3}$ was located at least 100 m above $Z_T$ (hereafter, separated profiles). About 50 % of the in situ cloud sampling

across the three IOPs was conducted during contact profiles (Table 1). Due to inter-annual variability, contact profiles accounted for about 42 %, 91 %, and 39 % of the in situ cloud sampling during the 2016, 2017, and 2018 IOPs, respectively.

The average $N_c$ and $R_e$ for all cloud profiles across the three IOPs were 157 ± 96 cm$^{-3}$ and 8.2 ± 2.7 μm, respectively (Table 3). The high proportion of contact profiles during the 2017 IOP

was associated with higher average $N_c$ and lower average $R_e$ (229 cm$^{-3}$ and 6.9 μm) compared to the 2016 IOP (150 cm$^{-3}$ and 7.0 μm) and the 2018 IOP (132 cm$^{-3}$ and 9.8 μm). It is possible that the use of CDP-B data for the 2017 IOP contributed to the increase in average $N_c$ relative to the 2016 IOP. However, the difference between the average CAS $N_c$ and the average CDP-B $N_c$ for the 2017 IOP (12 cm$^{-3}$) was lower than the difference between the average $N_c$ for the 2016 and 2017

IOPs (79 cm$^{-3}$). The difference between the $N_c$ for these IOPs were thus primarily due to the conditions at the cloud sampling locations. The microphysical differences between the 2016 and 2017 IOPs were associated with differences in surface precipitation. Based on the W-band retrievals from the Jet Propulsion Laboratory Airborne Precipitation Radar Version 3, the 2017 IOP had fewer profiles with precipitation reaching the surface (13 %) compared to the 2016 IOP

(34 %) (Dzambo et al., 2019).

On average, contact profiles had significantly higher $N_c$ (95 % CIs: 84 to 90 cm$^{-3}$ higher) and lower $R_e$ (95 % CIs: 1.4 to 1.6 μm lower) compared to separated profiles (throughout the



study, the term "significant" is exclusively used to represent statistical significance). The significant differences in $N_c$ and $R_e$ were associated with significantly higher $\tau$ (95 % CIs: 0.04 to

3.06 higher) for contact profiles, in accordance with the Twomey effect (Twomey, 1974; 1977). These results were consistent with the 2016 IOP when the contact profiles had higher $N_c$ (95 % CIs: 60 to 68 cm$^{-3}$ higher), lower $R_e$ (95 % CIs: 1.1 to 1.3 $\mu$m lower), and higher $\tau$ (95 % CIs: 1.1 to 4.3 higher) (G21).

        The median $N_c$ increased as a function of normalized height above cloud base ($Z_N$) for $Z_N$

$\leq 0.25$ consistent with droplet nucleation (Figure 3a). The median $N_c$ decreased near cloud top for $Z_N \geq 0.75$ from 204 to 154 cm$^{-3}$ for contact and from 104 to 69 cm$^{-3}$ for separated profiles. This was consistent with droplet evaporation associated with cloud-top entrainment (G21). The median $R_e$ increased with $Z_N$ consistent with condensational growth (Figure 3b). There was a greater increase in the median $R_e$ from cloud base to cloud top for separated profiles (from 7.1

to 9.5 $\mu$m) compared to contact profiles (from 6.1 to 7.9 $\mu$m). This is consistent with previous observations of stronger droplet growth in cleaner conditions as a function of $Z_N$ (Braun et al., 2018; G21) and LWP (Rao et al., 2020). Statistically insignificant differences between the average $H$ for contact and separated profiles suggest that the differential droplet growth was associated differences in cloud processes like collision-coalescence (further discussed in Section 5).

Eq. (5) shows the relative dependence of $\tau_{ad}$ on $N_c$ and LWP. The LWC and LWP responses to changes in aerosol conditions were examined because the adiabatic model suggests $\tau \alpha \text{LWP}^{5/6}$ (Eq. 5) (Brenguier et al., 2000). Contact profiles had significantly higher LWC, but the relative increase was less than 10 % (Table 4). It is possible this represents the lower limit of the aerosol



influence on cloud water since the aerosol influence varies with droplet size, precipitation

formation processes (Section 5), and the buffering by meteorological conditions (Section 7).

LWC was divided into rainwater content (RWC) and cloud water content (CWC) based on

droplet size. Droplets with $D > 50$ µm were defined as drizzle (Abel and Boutle, 2012; Boutle et

al., 2014) and the total drizzle mass was defined as RWC. The droplet mass for $D < 50$ µm was

defined as CWC. RWP and CWP were defined as the vertical integrals of RWC and CWC,

respectively. The median CWC increased with $Z_N$ but decreased over the top 10 % of the cloud

layer for contact profiles and over the top 20 % of the cloud layer for separated profiles consistent

with cloud-top entrainment (Figure 3c). For contact profiles, the median RWC increased with $Z_N$

before decreasing for $Z_N \geq 0.75$. The median RWC for separated profiles varied with $Z_N$. The

bottom half of the cloud layer had higher median values (up to 8.7 x $10^{-3}$ g m$^{-3}$) compared to the

top half (up to 7.0 x $10^{-3}$ g m$^{-3}$) (Figure 3d).

For contact profiles, there was a significant increase in the average CWC (10 %) and a

significant decrease in the average RWC (60 %) compared to separated profiles (Table 4). Contact

profiles also had significantly lower average RWP with insignificant differences for average CWP

(Table 4). Contact profiles were located in deeper boundary layers with significantly higher $Z_B$ and

$Z_T$ compared to separated profiles. However, the decrease in RWC cannot be attributed to

differences in $H$ or LWP (Kubar et al., 2009) because of statistically similar $H$ and LWP for contact

and separated profiles, on average (Table 4). These results show that instances of contact

between above-cloud aerosols and the MSC were associated with more numerous and smaller



cloud droplets and weaker droplet growth compared to instances of separation between the

above-cloud aerosols and the MSC.

**5 Precipitation Formation and *H***

Precipitation rate $R_p$ was calculated using the drizzle water content and fall velocity $u(D)$

following Abel and Boutle (2012),

$$R_p = \pi/6 \ \int_{50 \ \mu m}^{\infty} n(D)D^3 u(D)dD \qquad (6)$$

with fall velocity relationships from Rogers and Yau (1989) used in the computation.

Contact profiles had significantly lower $R_p$ compared to separated profiles (95 % CIs: 0.03

to 0.05 mm h$^{-1}$ lower). This suggests contact between the MSC and above-cloud biomass burning

aerosols was associated with precipitation suppression. LWP and *H* impact the sign and

magnitude of the precipitation changes in response to changes in aerosol conditions (Kubar et

al., 2009; Christensen and Stephens, 2012). Thus, cloud and precipitation properties were

evaluated as a function of *H* to examine the aerosol-induced changes in precipitation formation.

The 95$^{th}$ percentile was used to represent the maximum value of a variable. For example,

the 95$^{th}$ percentile of $R_p$ (denoted by $R_{p95}$) represents the maximum $R_p$ during a cloud profile.

Although more numerous contact profiles were drizzling compared to separated profiles, the

latter had more numerous profiles with high precipitation intensity. For instance, 114 out of 173

contact and 95 out of 156 separated profiles were drizzling with $R_{p95} > 0.01$ mm h$^{-1}$, out of which

contact and 40 separated profiles had $R_{p95} > 0.1$ mm h$^{-1}$, and only 1 contact and 9 separated


profiles had $R_{p95} > 1$ mm h$^{-1}$ (Figure 4a). This is consistent with radar retrievals of surface $R_p < 1$ mm h$^{-1}$ for over 93 % of the radar profiles from 2016 and 2017 (Dzambo et al., 2019).

**5.1. Microphysical properties**

On average, separated profiles had greater $R_{p95}$ (0.22 mm h$^{-1}$) compared to contact profiles (0.07 mm h$^{-1}$). $R_{p95}$ was positively correlated with $H$ as thicker profiles had higher precipitation intensity (Figure 4a). The average $R_{p95}$ increased from thin ($H < 175$ m) to thick clouds ($H > 175$ m) from 0.04 to 0.10 mm h$^{-1}$ for contact and 0.13 to 0.29 mm h$^{-1}$ for separated

profiles. Precipitation intensity thus decreased from separated to contact profiles for both thin and thick profiles. The average $R_{p95}$ for thin and thick contact profiles were 32 % and 37 % of the average $R_{p95}$ for thin and thick separated profiles, respectively.

CWC$_{95}$ was positively correlated with $H$ as thicker clouds had higher droplet mass (Figure 4b). This was consistent with condensational and collision-coalescence growth continuing to

occur with greater height above cloud base (Figure 3b, c), and greater cloud depth allowing for greater droplet growth. $N_{c95}$ and $R_{e95}$ were negatively and positively correlated with $H$, respectively (Figure 4c, d). The trends in $N_c$ and $R_e$ versus $H$ were consistent with the process of collision-coalescence resulting in fewer and larger droplets.

On average, contact profiles had higher $N_{c95}$ and lower $R_{e95}$ (311 cm$^{-3}$ and 8.6 μm)

compared to separated profiles (166 cm$^{-3}$ and 10.8 μm). It can be inferred that the presence of more numerous and smaller droplets during contact profiles decreased the efficiency of collision-coalescence. Alternatively, there may not have been sufficient time during the ascent to produce the few large droplets needed to broaden the size distribution and initiate collision-coalescence.



Since contact and separated profiles had statistically similar $H$ (Table 4), the following discussion

examines the link between precipitation suppression and the aerosol-induced changes in $N_c$, $R_e$,

and $LWC$ and their impact on precipitation.

### 5.2. Precipitation properties

Precipitation formation process rates were estimated using equations used in numerical

models to compare precipitation formation between contact and separated profiles.

Precipitation development in models is explained using bulk microphysical schemes. GCMs or LES

models parameterize precipitation formation using $S_{AUTO}$ and $S_{ACC}$ (e.g., Penner et al., 2006;

Morrison and Gettelman, 2008; Gordon et al., 2018). The most commonly used

parameterizations were used to estimate equivalent rates of precipitation formation from

models. $S_{AUTO}$ and $S_{ACC}$ were calculated following Khairoutdinov and Kogan (2000),

$$S_{AUTO} = (dw_r)_{AUTO} /dt = 1350 \ w_c^{2.47} N_c^{-1.79} \tag{7}$$

and

$$S_{ACC} = (dw_r)_{ACC}/dt = 67 \ (w_c w_r)^{1.15} \tag{8}$$

where $w_c$ and $w_r$ are cloud water and rainwater mixing ratios, respectively, and equal to the $CWC$

and $RWC$ divided by $\rho_a$.

Contact profiles had significantly lower $S_{AUTO}$ and $S_{ACC}$ compared to separated profiles (Table 4).

This is consistent with significantly lower RWC and $R_p$ for contact profiles and the association of

$S_{AUTO}$ and $S_{ACC}$ with precipitation onset and precipitation intensity, respectively. $S_{AUTO95}$ and $S_{ACC95}$

were positively correlated with $H$ (Figure 5a, b). Separated profiles had higher $S_{AUTO95}$ and $S_{ACC95}$



(9.6 x $10^{-10}$ s$^{-1}$ and 2.2 x $10^{-8}$ s$^{-1}$) compared to contact profiles (2.9 x $10^{-10}$ s$^{-1}$ and 1.2 x $10^{-8}$ s$^{-1}$)

associated with the inverse relationship between $S_{AUTO}$ and $N_c$ (Eq. 7). Faster autoconversion

resulted in higher drizzle water content and greater accretion of droplets on drizzle drops.

The sampling of lower $N_{c95}$ and higher $R_{e95}$ compared to thinner profiles suggests that

collision-coalescence was more effective in profiles with higher $H$ (Figure 4c, d). Thin contact

profiles had the lowest $S_{AUTO95}$ (1.4 x $10^{-10}$ s$^{-1}$) followed by thick contact (4.5 x $10^{-10}$ s$^{-1}$), thin

separated (4.7 x $10^{-10}$ s$^{-1}$), and thick separated profiles (1.4 x $10^{-9}$ s$^{-1}$). High $N_c$ and low CWC for

thin contact profiles (Figure 4b, c) are consistent with increased competition for cloud water

leading to weaker autoconversion. It is hypothesized that these microphysical differences

resulted in the lower $S_{AUTO95}$ and $R_{p95}$ for thin contact profiles compared to other profiles. The

differences between $R_p$ for contact and separated profiles thus varied with $H$ in addition to $N_c$,

$R_e$, and CWC. $N_c$, $R_e$, and CWC varied with $N_a$ (Section 4) and ACIs were examined in Sections 6

and 7.

**6 Aerosol Influence on Precipitation**

**6.1. Below-cloud $N_a$**

Polluted boundary layers in the southeast Atlantic are associated with entrainment

mixing between the free troposphere and the boundary layer (Diamond et al., 2018). For the

2016 IOP, contact profiles were located in boundary layers with significantly higher $N_a$ (95 % *CIs*:

to 115 cm$^{-3}$ higher) and carbon monoxide (CO) (95 % *CIs*: 13 to 16 ppb higher) compared to

separated profiles (G21). This is consistent with data from all three IOPs when contact profiles





were located in boundary layers with higher $N_a$ (95 % CIs: 231 to 249 cm$^{-3}$ higher) and CO (95 %

CIs: 27 to 29 ppb higher).

Following G21, 171 contact and 148 separated profiles from the IOPs were classified into

four regimes, Contact, high $N_a$ (C-H), Contact, low $N_a$ (C-L), Separated, high $N_a$ (S-H), and

Separated, low $N_a$ (S-L), where "low $N_a$" meant the profile was in a boundary layer with $N_a < 350$

cm$^{-3}$ up to 100 m below cloud base. Boundary layer CO concentration above 100 ppb was

sampled during 107 contact and 31 separated profiles, respectively. Contact profiles were more

often located in high $N_a$ boundary layers (131 out of 171 profiles classified as C-H) while separated

profiles were more often located in low $N_a$ boundary layers (108 out of 148 profiles classified as

S-L). This suggests contact between MSC cloud tops and above-cloud biomass burning aerosols

was associated with the entrainment of biomass-burning aerosols into the boundary layer.

Contact profiles had significantly higher $N_c$ and significantly lower $R_e$ relative to separated

profiles in both high $N_a$ (C-H relative to S-H) and low $N_a$ (C-L relative to S-L) boundary layers

(Figure 6). This was associated with significantly higher above- and below-cloud $N_a$ for the contact

profiles (Table 5). The differences in $N_c$ and $R_e$ were higher in high $N_a$ boundary layers where the

differences in above- and below-cloud $N_a$ were also higher compared to low $N_a$ boundary layers

(Table 5). This was consistent with previous observations of MSC cloud properties (Diamond et

al., 2018; Mardi et al., 2019) and similar analysis for data from the 2016 IOP (G21).

C-L profiles had significantly higher $N_c$ (95 % CIs: 5 to 14 cm$^{-3}$ higher) compared to S-H

profiles despite having significantly lower below-cloud $N_a$ (95 % CIs: 69 to 85 cm$^{-3}$ lower).

Significantly higher above-cloud $N_a$ for C-L profiles (95 % CIs: 321 to 361 cm$^{-3}$ higher) suggests





that this was associated with the influence of above-cloud $N_a$ on $N_c$. However, the smaller

difference in $N_c$ compared to the differences between C-H or S-H or C-L and S-L profiles suggests

the combined impact of above- and below-cloud $N_a$ was stronger than the impact of above-cloud

$N_a$ alone. These comparisons were qualitatively consistent when thresholds of 300 cm$^{-3}$ or 400

cm$^{-3}$ were used to define a low $N_a$ boundary layer.

**6.2. $N_c$ and $R_p$ versus $H$**

The cloud profiles were divided into four populations based on $H$ to compare $N_c$ and $R_p$

between different aerosols conditions while $H$ was constrained. The populations were defined

using the quartiles of $H$ (129, 175, and 256 m) to ensure similar sample sizes (Table 6). For each

population, contact profiles had higher $N_c$ and lower $R_p$ (Figure 7a, b) consistent with comparisons

averaged over all profiles (Table 4). The average $N_c$ decreased and the average $R_p$ increased with

$H$ (Figure 7a, b). For contact profiles, the average $N_c$ decreased with $H$ from 221 to 191 cm$^{-3}$ and

the average $R_p$ increased from 0.03 to 0.07 mm h$^{-1}$. For separated profiles, the average $N_c$

decreased from 149 to 92 cm$^{-3}$ and the average $R_p$ increased from 0.06 to 0.21 mm h$^{-1}$ over the

same range of $H$. These trends show the impact of collision-coalescence with increasing $H$.

For C-H profiles, high above- and below-cloud $N_a$ were associated with the highest

average $N_c$ and the lowest average $R_p$ among the four regimes (Figure 7c, d). C-H profiles had the

smallest increase in the average $R_p$ with $H$ (0.02 to 0.04 mm h$^{-1}$). Conversely, for S-L profiles, low

above- and below-cloud $N_a$ were associated with the lowest average $N_c$, the highest average $R_p$,

and the highest increase in the average $R_p$ with $H$ (0.12 to 0.29 mm h$^{-1}$). For each regime, the

average $N_c$ decreased with $H$ (except C-L) and the average $R_p$ increased with $H$ (Figure 7c, d).



### 6.3. Precipitation Susceptibility $S_o$

$S_o$ was used to evaluate the dependence of $R_p$ on $N_c$ under the different aerosol conditions. $S_o$, defined as the negative slope between the natural logarithms of $R_p$ and $N_c$ (Feingold and Seibert, 2009), is given by

$$S_o = -\,d\ln(R_p)/d\ln(N_c)\,, \tag{9}$$


where a positive value indicates decreasing $R_p$ with increasing $N_c$, in accordance with the "lifetime effect" (Albrecht, 1989). The average $S_o$ across all profiles was 0.88 ± 0.03. On average, contact profiles had lower $S_o$ (0.87 ± 0.04) compared to separated profiles (1.08 ± 0.04). This was consistent with the hypothesis of lower values for $S_o$ analogues (where $N_c$ in Eq. (9) is replaced by $N_a$) in the presence of above-cloud aerosols (Duong et al., 2011). Modelling studies (Wood et


al., 2009; Jiang et al., 2010) have found $S_o$ depends on the ratio of $S_{ACC}$ to $S_{AUTO}$. $S_{ACC}$ is independent of $N_c$ (Eq. 8) and greater values of $S_{ACC}/S_{AUTO}$ represent a weaker dependence of $R_p$ on $N_c$. Lower $S_o$ for contact profiles was associated with higher $S_{ACC}/S_{AUTO}$ compared to separated profiles (Table 4).


$S_o$ was calculated as a function of $H$ (Figure 8) using $N_c$ and $R_p$ for the four populations of cloud profiles (Figure 9). The sensitivity of $S_o$ to the number of populations is discussed in Appendix A. Averaged over all profiles, $S_o$ had minor variations with $H$ (e.g., 0.67, 0.68, and 0.54 as $H$ increased) before increasing to 1.13 for $H > 256$ m (Table 6). This trend in $S_o$ versus $H$ was consistent with previous analyses of $S_o$ (Sorooshian et al., 2009; Jung et al., 2016). However,


different trends emerged when $S_o$ was calculated for contact and separated profiles.


The difference between $S_o$ for contact and separated profiles varied with $H$ and thin clouds ($H < 129$ m) had the highest difference. 30 separated profiles with $H < 129$ m had high $S_o$ (1.47 ± 0.10). This was because of strong dependence of $R_p$ on $N_c$ associated with higher average $R_p$ for low $N_c$ (< 100 cm$^{-3}$) measurements (0.18 mm h$^{-1}$) compared to high $N_c$ measurements (0.01

mm h$^{-1}$) (Figure 9a). In contrast, precipitation suppression and weaker droplet growth for thin contact profiles (Section 5) resulted in average $R_p < 0.03$ mm h$^{-1}$ for both low $N_c$ and high $N_c$ measurements (Fig. 9a). Thus, there was poor (and statistically insignificant) correlation between $N_c$ and $R_p$ ($R = -0.03$) which led to a low and statistically insignificant value for $S_o$ (-0.06 ± 0.11).

For separated profiles, $S_o$ decreased with $H$ to 0.53 ± 0.09 for $129 < H < 175$ m and to 0.34

± 0.07 for $175 < H < 256$ m (Figure 8a). This was because the average $R_p$ for the high $N_c$ measurements increased with $H$ from 0.01 mm h$^{-1}$ for thin profiles to 0.05 and 0.04 mm h$^{-1}$, respectively (Figure 9b, c). This was consistent with collision-coalescence beginning to occur for high $N_c$ measurements as droplet mass increased with $H$ (Figure 5b). $S_o$ increased to 1.45 ± 0.07 for the cloud population with $H > 256$ m. This population had lower $N_c$ and higher $R_p$ compared

to the populations with lower $H$ (Figure 7a, b). The average $R_p$ for low $N_c$ measurements (0.26 mm h$^{-1}$) was higher than high $N_c$ measurements (0.13 mm h$^{-1}$) (Figure 9d). These observations were consistent with collision-coalescence and stronger precipitation formation for low $N_c$ measurements. The latter was associated with the inverse relationship between $N_c$ and $S_{AUTO}$.

Contact profiles with $H > 129$ m had a significant correlation between $N_c$ and $R_p$. The

average $R_p$ increased with $H$ with a larger increase for the low $N_c$ measurements (0.028 to 0.12 mm h$^{-1}$) compared to the high $N_c$ measurements (0.03 to 0.06 mm h$^{-1}$). It is hypothesized that





collision-coalescence was hindered by the presence of more numerous droplets during the high $N_c$ measurements, and as droplet growth and collision-coalescence occurred with increasing $H$, the limiting factor for $R_p$ changed from $H$ to $N_c$. The dependence of $R_p$ on $N_c$ increased with $H$ and

as a result, $S_o$ increased with $H$ from 0.88 ± 0.06 to 1.15 ± 0.06 (Figure 8a).

S-L profiles had the highest $S_o$ (1.12) among the four regimes defined based on the above- and below-cloud $N_a$ (Table 7). This was associated with the S-L profiles having the lowest average $N_c$ and the highest average $R_p$ among the four regimes (Figure 7c, d). In descending order of $S_o$, S-L profiles were followed by C-L (0.86), S-H (0.50), and C-H profiles (0.33). Profiles in low $N_a$

boundary layers (S-L and C-L) had higher $S_o$ compared to profiles in high $N_a$ boundary layers (S-H and C-H). This was consistent with wet scavenging of below-cloud aerosols (Duong et al., 2011; Jung et al., 2016).

The sensitivity of $S_o$ to the inclusion of precipitating clouds is examined in Appendix B. C-L and C-H profiles had similar trends in $S_o$ except for the thinnest profiles ($H < 129$ m) (Figure 8b).

C-L profiles had an insignificant value for $S_o$ due to low sample size (4) and C-H profiles had negative $S_o$. These were thin profiles with little cloud water (Figure 5b), high $N_c$ (Figure 7c), and low $R_p$ (Figure 7d). It is hypothesized that increasing $N_c$ would provide the cloud water required for precipitation initiation and aid collision-coalescence.

out of 148 separated profiles were classified as S-L profiles. As a result, separated and

S-L profiles had similar trends in $S_o$ versus $H$ (Figure 8). On average, S-L profiles had higher $S_o$ than S-H profiles which could be associated with wet scavenging resulting in the lower below-cloud $N_a$ for S-L profiles. For S-H profiles, $S_o$ was constant with $H$ at about 0.45 (except for the population


with 175 < $H$ < 256 m which had an insignificant value for $S_o$) (Table 7). The $S_o$ comparisons

between profiles located in high $N_a$ or low $N_a$ boundary layers varied with the sample sizes of the

populations. The sample sizes varied based on the threshold used to define a low $N_a$ boundary

layer which is discussed in Appendix C.

**6.4. $S_o$ Discussion**

Higher $N_c$ and lower $R_e$ for contact profiles led to precipitation suppression and lower

$S_{AUTO}$, $S_{ACC}$, and $R_p$ which were associated with lower $S_o$ compared to separated profiles. Polluted

clouds were thus less susceptible to precipitation suppression than cleaner clouds. The

differences in $S_o$ varied with $H$ due to the variability in $R_p$, $N_c$, $R_e$, and CWC associated with aerosols

and droplet growth. The change in $S_o$ was highest for thin polluted clouds due to poor correlation

between $N_c$ and $R_p$ as limited droplet growth led to low $R_p$ regardless of the $N_c$. Power-law

relationships between $R_p$, $N_c$, and $H$ (Geoffroy et al., 2008) thus need to account for changes in

the dependence of $R_p$ on $N_c/H$ associated with ACIs and $H$.

The trends in $S_o$ were only compared with studies analyzing airborne data due to the

variability in $S_o$ depending on whether aircraft, remote sensing, or modeling data were examined

(Sorooshian et al., 2019). Consistent with Terai et al. (2012), $S_o$ decreased with $H$ for separated

profiles with $H$ < 256 m. The results from Section 5 suggest droplet growth with $H$ decreased the

susceptibility to aerosols because $R_p$ was limited by droplet growth instead of $N_a$ or $N_c$. In

comparison, $S_o$ increased with $H$ for contact profiles consistent with Jung et al. (2016). The low $S_o$

for thin contact profiles was consistent with the low $S_o$ (0.06) for thin MSC over the southeast



Pacific (Jung et al., 2016). This was attributed to insufficient cloud water for precipitation initiation (as noted in Section 5).

Jung et al. (2016) analyzed MSC sampled farther east and away from South America compared to Terai et al. (2012). They argued a westward increase in precipitation frequency and intensity, along with a decrease in aerosols and $N_c$, led to the differences between the two studies. This same attribution on the role of aerosols can be made for the ORACLES data as there were differences between contact and separated profiles because the MSC sampled during these

profiles were located in similar geographical locations with different aerosol conditions. Modeling studies (e.g., Wood et al., 2009; Gettelman et al., 2013) have shown that $S_o$ increases with $H$ when $S_{AUTO}$ dominates $S_{ACC}$ (typically for $R_e < 14$ μm, the critical radius for precipitation initiation). Maximum $R_e < 14$ μm was sampled during all but 23 separated and 3 contact profiles (Figure 5d). This would explain the increase in $S_o$ with $H$ for both contact (for $H > 129$ m) and

separated profiles (for $H > 256$ m).

**7 Meteorological Influence on LWP**

The relationships between LWP or $H$ and $N_c$, $R_e$, and LWC depend on meteorological conditions in addition to aerosol properties. The MSC LWP and cloud cover can vary with LTS (Klein and Hartmann, 1993; Mauger and Norris, 2007), estimated inversion strength (EIS) (Wood

and Bretherton, 2006), and SST (Wilcox, 2010; Sakaeda et al., 2011). The correlations between LWP/$H$ and these parameters are examined using the European Centre for Medium-Range Weather Forecasts (ECMWF) atmospheric reanalysis (ERA5) (Hersbach et al., 2020) to define the meteorological conditions.



ERA5 provides hourly output with a horizontal resolution of 0.25° x 0.25° for 37 pressure

(p) levels (up to 1 hPa). The cloud sampling for most flights was conducted within three hours of

12:00 UTC (Table 2). ERA5 data at 12:00 UTC were thus used for the grid box nearest to the profile

(Dzambo et al., 2019). The low cloud cover (LCC), SST, $H_{BL}$, total column liquid water (ERA5 LWP)

and rainwater (ERA5 RWP), mean sea level pressure ($p_o$), 2 m temperature ($T_o$), and 2 m dew

point temperature ($T_d$) were examined (Table 8).

The difference between potential temperatures at 700 hPa and the surface was defined

as LTS (Klein and Hartmann, 1993). The lifting condensation level (LCL) was defined as LCL = 125

m K$^{-1}$ ($T_o$ - $T_d$) (Lawrence, 2005). EIS was calculated following Wood and Bretherton (2006),

$$EIS = LTS - \Gamma_m^{850}(z_{700} - LCL), \ \Gamma_m^{850} = \Gamma_m([T_o + T_{700}]/2, 850 \ mb), \qquad (10)$$

where $\Gamma_m$ is the moist adiabatic potential temperature gradient and $z_{700}$ is the height at 700 mb.

$\Gamma_m$ was calculated as

$$\Gamma_m(T, p) = \frac{g}{c_p} \left[ 1 - \frac{1 + L_v w_s(T,p)/R_a T}{1 + L_v^2 w_s(T,p)/c_p R_v T^2} \right], \qquad (11)$$

where $g$ is the gravitational acceleration, $c_p$ is the specific heat of air at constant pressure, $L_v$ is

the latent heat of vaporization, and $R_a$ and $R_v$ are the gas constants for dry air and water vapor,

respectively (Wood and Bretherton, 2006).

LCC refers to cloud fraction for $p > 0.8 \ p_o$, corresponding to $p > 810$ hPa, where most

profiles were sampled (Table 2). The ECMWF model used a threshold of EIS > 7 K to distinguish

between well-mixed boundary layers topped by stratocumulus and decoupled boundary layers



with cumulus clouds (ECMWF IFS Documentation, 2016). This distinction improved the agreement between the model LCC and LWP and observations (Köhler et al., 2011).

LCC was proportional to EIS/LTS, and LCC < 0.8 was mostly observed for EIS < 7 K (Figure 10a). Decoupled boundary layers can be topped by MSC (G21; Wood, 2012). Profiles with EIS < 7 K were included in the analysis if ERA5 had LCC > 0.95. This included 64 contact and 88 separated profiles from the three IOPs. For the 2016, 2017, and 2018 IOPs, 50, 20, and 76 profiles, respectively, had LCC > 0.95 out of which, 0, 4, and 44 profiles, respectively, had EIS < 7 K. The

average ERA5 $H_{BL}$ (599 ± 144 m) was lower than the average $Z_T$ (932 ± 196 m). This underestimation of $H_{BL}$ by ERA5 has been observed for stratocumulus over the southeast and northeast Pacific (Ahlgrimm et al., 2009; Hannay et al., 2009).

On average, the ERA5 LWP (51 ± 21 g m$^{-2}$) was slightly greater than LWP (46 ± 41 g m$^{-2}$), but the differences were statistically insignificant. There was a significant but weak correlation

between LWP and ERA5 LWP ($R$ = 0.18) (Figure 10b). On average, the ERA5 RWP (0.48 ± 1.07 g m$^{-2}$) was lower than RWP (1.19 ± 2.76 g m$^{-2}$). There were insignificant differences between ERA5 LWP/LWP for contact and separated profiles with LCC > 0.95 (Table 8). Contact profiles with LCC > 0.95 had significantly higher ERA5 RWP (Table 8). While this is counter-intuitive, given the precipitation suppression, it was due to selection of profiles with LCC > 0.95. Contact profiles with

LCC > 0.95 also had higher in situ RWP (95 % CIs: 0.32 to 2.08 g m$^{-2}$ higher) compared to separated profiles with LCC > 0.95.

LWP was positively correlated with SST and $T_o$ and negatively correlated with LTS and EIS with weak but statistically significant correlations (Figure 11). On average, separated profiles had



significantly higher SST (95 % CIs: 0.01 to 1.48 K higher) compared to contact profiles with

insignificant differences between the average $T_o$, EIS, and LTS. Since the correlation between

LWP/$H$ and SST was weak, it is unlikely the differences between contact and separated profiles

were driven by SST differences alone. When all profiles (irrespective of LCC) were considered,

there were insignificant differences between the average ERA5 RWP, SST, $T_o$, EIS, and LTS for

contact and separated profiles. This suggests the differences between contact and separated

profiles found during the ORACLES IOPs were primarily associated with ACIs instead of

meteorological effects.

**8 Conclusions**

In situ measurements of stratocumulus over the southeast Atlantic Ocean were collected

during the NASA ORACLES field campaign. The microphysical ($N_c$ and $R_e$), macrophysical (LWP and

$H$), and precipitation properties ($R_p$ and $S_o$) of the stratocumulus were analyzed. 173 "contact"

profiles with $N_a > 500$ cm$^{-3}$ within 100 m above cloud tops were compared with 156 "separated"

profiles with $N_a < 500$ cm$^{-3}$ up to at least 100 m above cloud tops. Contact between above-cloud

aerosols and the stratocumulus was associated with,

1.  More numerous and smaller droplets with weaker droplet growth with height.

Contact profiles had significantly higher $N_c$ (84 to 90 cm$^{-3}$ higher) and lower $R_e$ (1.4 to 1.6

µm lower) compared to separated profiles. The median $R_e$ had a smaller increase from cloud base

to cloud top for contact (6.1 to 7.9 µm) compared to separated profiles (7.1 to 9.5 µm). The

profiles had similar LWP and $H$, and it is hypothesized the differences in droplet growth were

associated with collision-coalescence.



2. The entrainment of above-cloud biomass-burning aerosols into the boundary layer and aerosol-induced cloud microphysical changes in both clean and polluted boundary layers.

Contact profiles were more often located in polluted boundary layers and had higher below-cloud CO concentration (27 to 29 ppb higher). Contact profiles had 25 to 31. $cm^{-3}$ higher $N_c$ and 0.2 to 0.5 μm lower $R_e$ in clean and 98 to 108 $cm^{-3}$ higher $N_c$ and 1.6 to 1.8 μm lower $R_e$ in polluted boundary layers.

3. Precipitation suppression with significantly lower precipitation intensity and precipitation formation process rates.

Separated profiles had $R_p$ up to 0.22 mm $h^{-1}$ while contact profiles had $R_p$ up to 0.07 mm $h^{-1}$. $S_{AUTO}$ and $S_{ACC}$ had higher maxima for separated (up to 9.6 x $10^{-10}$ $s^{-1}$ and 2.2 x $10^{-8}$ $s^{-1}$) compared to contact profiles (up to 2.9 x $10^{-10}$ $s^{-1}$ and 1.2 x $10^{-8}$ $s^{-1}$).

4. Lower precipitation susceptibility with the strongest impact in thin clouds ($H$ < 129 m).

Contact profiles had lower $S_o$ (0.87 ± 0.04) compared to separated profiles (1.08 ± 0.04). Thin clouds had the highest difference in $S_o$ (-0.06 ± 0.11 for contact and 1.47 ± 0.10 for separated). Lower $S_o$ for thin contact profiles was associated with poor correlation between $N_c$ and $R_p$ ($R$ = -0.03). For separated profiles, $S_o$ decreased with $H$ before increasing for $H$ > 256 m. In comparison, $S_o$ increased with $H$ for contact profiles for $H$ > 129 m.

5. Statistically insignificant differences in meteorological parameters that influence $LWP/H$.

Based on ERA5 reanalysis data, LWP was correlated with SST ($R$ = 0.22), $T_o$ ($R$ = 0.27), LTS ($R$ = - 0.29), and EIS ($R$ = - 0.31). Contact profiles with ERA5 LCC > 0.95 had lower SST (0.01 to 1.48



K lower) with similar $T_o$, LTS, and EIS compared to separated profiles. The SST differences were

insignificant when profiles with LCC < 0.95 were included in the comparison.

The ORACLES dataset addresses the "lack of long-term data sets needed to provide

statistical significance for a sufficiently large range of aerosol variability influencing specific cloud

regimes over a range of macrophysical conditions" (Sorooshian et al., 2010). Three important

factors affecting $S_o$ were discussed (Sorooshian et al., 2019): above-cloud $N_a$, below-cloud $N_a$, and

meteorological conditions. This study analyzed ORACLES data from all three IOPs and the first

two conclusions were consistent with the analysis of ORACLES 2016 (Gupta et al., 2021). Future

work will compare in situ data with $R_p$ retrievals from the Airborne Precipitation Radar (Dzambo

et al., 2021) to evaluate the sensitivity of $S_o$ to the use of satellite retrievals of $R_p$ (Bai et al., 2018).

Vertical profiles of MSC cloud properties will be used to evaluate satellite retrievals (Painemal

and Zuidema, 2011; Zhang and Platnick, 2011) to address the uncertainties associated with

satellite-based estimates of ACIs (Quaas et al., 2020).




**APPENDIX A – Sensitivity studies on dependence of $S_o$ on $H$**

The base analysis examined how cloud properties varied with $H$ by separating cloud profiles into four populations of $H$ using the following endpoints: 28, 129, 175, 256, and 700 m. Two sensitivity studies determine if trends describing the variation of $N_c$, $R_p$, and $S_o$ with $H$ were sensitive to the endpoints used to sort cloud profiles into different populations.

    First, cloud profiles were classified into two populations using the median $H$ (175 m) to
divide the populations (Table A1). The average $N_c$ decreased and the average $R_p$ increased with $H$ for both contact (211 to 186 cm$^{-3}$ and 0.03 to 0.07 mm h$^{-1}$) and separated profiles (129 to 104 cm$^{-3}$ and 0.07 to 0.15 mm h$^{-1}$). $S_o$ increased with $H$ for contact profiles from 0.53 to 1.06 and slightly decreased with $H$ for separated profiles from 1.05 to 1.02 (Table A1). The difference between $S_o$ for contact and separated profiles was greater for thin profiles ($H < 175$ m) compared
to thick profiles ($H > 175$ m). These results are consistent with trends using four populations but provide less detail about how $S_o$ varies with $H$ (Fig. A1).

    Second, cloud profiles were classified into three populations using the terciles of $H$ (145 and 224 m) (Table A1). The average $N_c$ decreased and the average $R_p$ increased from the lowest to the highest $H$ for contact (231 to 187 cm$^{-3}$ and 0.03 to 0.07 mm h$^{-1}$) and separated profiles (138
to 95 cm$^{-3}$ and 0.06 to 0.18 mm h$^{-1}$). For separated profiles, $S_o$ first decreased with $H$ from 1.15 to 0.25 before increasing to 1.45 for the highest $H$ (Fig. A1). Contact profiles had insignificant $S_o$ for the lowest $H$ followed by $S_o$ increasing from 0.95 to 1.08 with $H$. The results presented here are robust as relates to the number of populations used.





**APPENDIX B – Sensitivity studies on dependence of $S_o$ on $R_p$**

Another sensitivity study examined the $R_p$ threshold used for cloud profiles included while calculating $S_o$. The average $S_o$ decreased if weakly precipitating clouds with low $R_p$ were excluded (Fig. B1, Table B1). It is possible that this was due to the higher $N_a$ and $N_c$ associated with weakly precipitating clouds. The exclusion of weakly-precipitating clouds provides biased trends in $S_o$

since these clouds could have undergone precipitation suppression already. Conversely, strongly precipitating clouds were associated with cleaner conditions and lower $N_a$ and $N_c$. The exclusion of strongly precipitating clouds also leads to a decrease in the average $S_o$ (Fig. B2, Table B1).

The occurrence of wet scavenging below strongly precipitating clouds (Duong et al., 2011) results in lower below-cloud $N_a$ (and subsequently $N_c$). Higher susceptibility to precipitation

suppression for cleaner, strongly precipitating clouds would explain the increase in the average $S_o$. This is consistent with observations of $S_o$ using different $R_p$ thresholds (c.f. Fig B1, Jung et al., 2016) and hypotheses regarding the impact of different $N_a$ on $S_o$ (Duong et al., 2011; Fig. 11, Jung et al., 2016).

**APPENDIX C – Dependence of $S_o$ on the definition of clean and polluted boundary layers**

The number of cloud profiles classified into the S-L, C-L, S-H, and C-H regimes varied depending on the below-cloud $N_a$ threshold used to define a low $N_a$ or clean boundary layer. For the threshold used in the base analysis (350 cm$^{-3}$), contact profiles were more often located in polluted boundary layers (131 out of 171 profiles classified as C-H) while separated profiles were more often located in clean boundary layers (108 out of 148 profiles classified as S-L). The

comparisons between $S_o$ in clean and polluted boundary layers varied with the threshold used.



As a sensitivity study, a lower threshold was used to define a clean boundary layer (300 cm$^{-3}$). For this case, the C-L regime had no profiles in the population with the lowest $H$ ($H$ < 129 m) when four populations of profiles were used to examine the dependence of $S_o$ on $H$. Two out of the other three populations had an insignificant value for $S_o$ due to poor and statistically

insignificant correlations between $N_c$ and $R_p$ (Table C1). This was associated with a low sample size for the populations (6 each). A second sensitivity study used a higher threshold to define a clean boundary layer (400 cm$^{-3}$). For this case, the S-H regime has insignificant $S_o$ for three out of the four populations of $H$ and the remaining population had a small sample size (3 profiles) (Table C1). The base analysis using a threshold of 350 cm$^{-3}$ to define a clean boundary layer was used to

compare $S_o$ values that represent a larger number of cloud profiles.





*Code availability.* University of Illinois/Oklahoma Optical Array Probe (OAP) Processing Software

is available at https://doi.org/10.5281/zenodo.1285969 (McFarquhar et al., 2018). The Airborne

Data    Processing    and    Analysis    software    package    is    available    at

https://zenodo.org/record/3733448 (Delene et al., 2020).

*Data availability*. All ORACLES data are accessible via the digital object identifiers provided under

ORACLES Science Team references: https://doi.org/10.5067/Suborbital/ORACLES/P3/2018_V2

(ORACLES Science Team, 2020a), https://doi.org/10.5067/Suborbital/ORACLES/P3/2017_V2

(ORACLES Science Team, 2020b), https://doi.org/10.5067/Suborbital/ORACLES/P3/2016_V2

(ORACLES Science Team, 2020c). ERA5 data were obtained from Climate Data Store (last access:

18 May 2021): https://cds.climate.copernicus.eu/cdsapp#!/home (Hersbach et al., 2020).

*Author contributions.* GMM and MRP worked with other investigators to design the ORACLES

project and flight campaigns. SG designed the study with guidance from GMM. SG analyzed the

data with inputs from GMM, JRO'B, and MRP. JRO'B and DJD processed PCASP data and cloud

probe data, conducted data quality tests, and some of the data comparisons between cloud

probes. SG processed 2D-S and HVPS-3 data and conducted some of the data comparisons

between cloud probes. JDSG processed PDI data. GMM and MRP acquired funding. All authors

were involved in data collection during ORACLES. SG wrote the manuscript with guidance from

GMM and reviews from all authors.

*Competing interests*. The authors declare that they have no conflicts of interest.



*Special issue statement.* This article is part of the special issue "New observations and related modeling studies of the aerosol–cloud–climate system in the Southeast Atlantic and southern

Africa regions (ACP/AMT inter-journal SI)". This article is not associated with a conference.

*Acknowledgements.* The authors thank Yohei Shinozuka for providing merged instrument data files for the ORACLES field campaign. We acknowledge the entire ORACLES science team for their assistance with data acquisition, analysis, and interpretation. We thank the NASA Ames Earth Science Project Office and the NASA P-3B flight/maintenance crew for logistical and aircraft

support. Some of the computing for this project was performed at the OU Supercomputing Center for Education & Research (OSCER) at the University of Oklahoma (OU).

*Financial support.* Funding for this project was obtained from NASA Award #80NSSC18K0222. ORACLES is funded by NASA Earth Venture Suborbital-2 grant NNH13ZDA001N-EVS2. SG was supported by NASA headquarters under the NASA Earth and Space Science Fellowship grants

NNX15AF93G and NNX16A018H. GMM and SG acknowledge support from NASA grant 80NSSC18K0222.







Table 1: The number of cloud profiles (n) for P-3 research flights (PRFs) analyzed in the study, number of contact and separated profiles with sampling time in parentheses, and instruments that provided valid samples of droplets with $D < 50$ μm (instrument used for analysis is in bold).

| PRF number and date | n | Contact | Separated | Instruments |
|---|---|---|---|---|
| PRF05Y16: Sep. 06 | 24 | 13 (857 s) | 11 (470 s) | **CAS**, PDI |
| PRF07Y16: Sep. 10 | 9 | 0 (0 s) | 9 (461 s) | **CAS**, PDI |
| PRF08Y16: Sep. 12 | 8 | 1 (32 s) | 7 (472 s) | **CAS**, PDI |
| PRF09Y16: Sep. 14 | 8 | 0 (0 s) | 8 (574 s) | **CAS**, PDI |
| PRF11Y16: Sep. 20 | 13 | 13 (669 s) | 0 (0 s) | **CAS**, PDI |
| PRF13Y16: Sep. 25 | 9 | 3 (148 s) | 6 (363 s) | **CAS**, PDI |
| PRF01Y17: Aug. 12 | 15 | 14 (499 s) | 1 (25 s) | CAS, **CDP-B** |
| PRF02Y17: Aug. 13 | 17 | 17 (754 s) | 0 (0 s) | CAS, **CDP-B** |
| PRF03Y17: Aug. 15 | 12 | 12 (272 s) | 0 (0 s) | CAS, **CDP-B** |
| PRF04Y17: Aug. 17 | 7 | 7 (127 s) | 0 (0 s) | CAS, **CDP-B** |
| PRF07Y17: Aug. 21 | 13 | 9 (188 s) | 4 (76 s) | CAS, **CDP-B** |
| PRF08Y17: Aug. 24 | 9 | 9 (324 s) | 0 (0 s) | CAS, **CDP-B** |
| PRF10Y17: Aug. 28 | 11 | 7 (496 s) | 4 (168 s) | CAS, **CDP-B** |
| PRF01Y18: Sep. 27 | 21 | 0 (0 s) | 21 (933 s) | CAS, **CDP-B**, CDP-C |
| PRF02Y18: Sep. 30 | 13 | 7 (337 s) | 6 (183 s) | CAS, **CDP-B**, CDP-C |
| PRF04Y18: Oct. 03 | 5 | 0 (0 s) | 5 (137 s) | CAS, **CDP-B**, CDP-C |
| PRF05Y18: Oct. 05 | 4 | 4 (109 s) | 0 (0 s) | CAS, **CDP-B**, CDP-C |
| PRF06Y18: Oct. 07 | 10 | 10 (337 s) | 0 (0 s) | CAS, **CDP-B**, CDP-C |
| PRF07Y18: Oct. 10 | 13 | 11 (472 s) | 2 (153 s) | **CDP-B**, CDP-C |
| PRF08Y18: Oct. 12 | 19 | 0 (0 s) | 19 (773 s) | CDP-B, **CDP-C** |
| PRF09Y18: Oct. 15 | 30 | 17 (766 s) | 13 (365 s) | **CDP-B**, CDP-C |
| PRF11Y18: Oct. 19 | 12 | 0 (0 s) | 12 (731 s) | CDP-B, **CDP-C** |
| PRF12Y18: Oct. 21 | 18 | 0 (0 s) | 18 (833 s) | CDP-B, **CDP-C** |
| PRF13Y18: Oct. 23 | 29 | 19 (777 s) | 10 (366 s) | CDP-B, **CDP-C** |
| **Total (2016)** | **71** | **30 (1,706 s)** | **41 (2,340 s)** | |
| **Total (2017)** | **84** | **75 (2,660 s)** | **9 (269 s)** | |
| **Total (2018)** | **174** | **68 (2,798 s)** | **106 (4,474 s)** | |
| **Total** | **329** | **173 (7,164 s)** | **156 (7,083 s)** | |







Table 2: Range of time, latitude, longitude, $Z_T$ and cloud top pressure ($P_T$) for PRFs in Table 1.

| PRF | Time (UTC) | Latitude (°S) | Longitude (°E) | $Z_T$ (m) | $P_T$ (mb) |
|---|---|---|---|---|---|
| PRF05Y16: Sep. 06 | 08:46 - 12:35 | 10.2 - 19.7 | 9.00 - 11.9 | 359 - 1002 | 904 - 976 |
| PRF07Y16: Sep. 10 | 09:09 - 12:36 | 14.1 - 18.7 | 4.00 - 8.60 | 990 - 1201 | 885 - 908 |
| PRF08Y16: Sep. 12 | 11:16 - 12:26 | 9.70 - 12.9 | -0.30 - 3.00 | 1146 - 1226 | 881 - 890 |
| PRF09Y16: Sep. 14 | 09:36 - 14:16 | 16.4 - 18.1 | 7.50 - 9.00 | 635 - 824 | 922 - 945 |
| PRF11Y16: Sep. 20 | 08:44 - 13:11 | 15.7 - 17.3 | 8.90 - 10.5 | 432 - 636 | 941 - 966 |
| PRF13Y16: Sep. 25 | 10:59 - 13:51 | 10.9 - 14.3 | 0.80 - 4.30 | 729 - 1124 | 890 - 934 |
| PRF01Y17: Aug. 12 | 11:30 - 15:01 | 2.41 - 13.0 | 4.84 - 5.13 | 748 - 1379 | 866 - 933 |
| PRF02Y17: Aug. 13 | 10:15 - 13:07 | 7.20 - 9.00 | 4.50 - 5.00 | 779 - 1384 | 865 - 928 |
| PRF03Y17: Aug. 15 | 11:26 - 13.32 | 9.08 - 15.0 | 4.96 - 5.00 | 536 - 1148 | 887 - 954 |
| PRF04Y17: Aug. 17 | 12:03 - 16:14 | 7.99 - 9.43 | -7.0 - -12.8 | 1547 - 1782 | 827 - 848 |
| PRF07Y17: Aug. 21 | 13:20 - 16:37 | 7.96 - 8.05 | -8.16 - 3.32 | 1061 - 1491 | 855 - 897 |
| PRF08Y17: Aug. 24 | 11:28 - 14:58 | 4.90 - 14.8 | 4.97 - 5.15 | 911 - 2015 | 801 - 916 |
| PRF10Y17: Aug. 28 | 11:46 - 13:18 | 7.84 - 11.0 | 4.89 - 5.01 | 1070 - 1216 | 881 - 897 |
| PRF01Y18: Sep. 27 | 10:07 - 13:11 | 5.66 - 12.1 | 4.87 - 5.03 | 819 - 1169 | 885 - 922 |
| PRF02Y18: Sep. 30 | 09:50 - 12:24 | 6.85 - 8.18 | 4.94 - 5.13 | 747 - 840 | 920 - 930 |
| PRF04Y18: Oct. 03 | 13:17 - 14:41 | -1.05 - 4.61 | 5.00 - 5.06 | 1137 - 2151 | 790 - 888 |
| PRF05Y18: Oct. 05 | 07:22 - 10:09 | 9.50 - 9.63 | 5.79 - 6.66 | 780 - 892 | 915 - 928 |
| PRF06Y18: Oct. 07 | 11:04 - 11:29 | 10.1 - 11.8 | 5.00 - 5.00 | 863 - 928 | 913 - 918 |
| PRF07Y18: Oct. 10 | 10:16 - 13:31 | 4.46 - 13.1 | 4.88 - 5.09 | 926 - 1329 | 866 - 912 |
| PRF08Y18: Oct. 12 | 13:02 - 16:19 | 1.02 - 4.58 | 5.50 - 6.96 | 1073 - 1905 | 813 - 895 |
| PRF09Y18: Oct. 15 | 10:27 - 13:09 | 5.25 - 14.1 | 4.91 - 5.00 | 693 - 1547 | 849 - 937 |
| PRF11Y18: Oct. 19 | 11:58 - 13:00 | 6.50 - 7.70 | 8.00 - 9.06 | 701 - 1276 | 873 - 932 |
| PRF12Y18: Oct. 21 | 10:21 - 13:07 | 4.91 - 13.5 | 4.88 - 5.00 | 675 - 983 | 902 - 936 |
| PRF13Y18: Oct. 23 | 10:28 - 13:38 | 3.07 - 5.00 | -2.65 - 5.00 | 873 - 1281 | 873 - 915 |





Table 3: Average values for cloud properties measured during cloud profiles from the PRFs listed in Table 1 for each *IOP*. Error estimates represent one standard deviation. *R* between *LWP* estimates and *H* in parentheses.

| Parameter | 2016 | 2017 | 2018 | All |
|---|---|---|---|---|
| Profile count | 71 | 84 | 174 | 329 |
| $N_c$ (cm$^{-3}$) | 150 ± 73 | 229 ± 108 | 132 ± 87 | 157 ± 96 |
| $R_e$ (µm) | 7.0 ± 1.9 | 6.9 ± 1.6 | 9.8 ± 3.3 | 8.2 ± 2.7 |
| LWC (g m$^{-3}$) | 0.15 ± 0.09 | 0.21 ± 0.15 | 0.26 ± 0.17 | 0.22 ± 0.16 |
| King LWC (g m$^{-3}$) | 0.29 ± 0.15 | 0.23 ± 0.17 | 0.24 ± 0.14 | 0.25 ± 0.15 |
| $\tau$ | 7.2 ± 3.6 | 7.2 ± 8.9 | 9.0 ± 7.7 | 8.8 ± 7.7 |
| H (m) | 244 ± 83 | 148 ± 92 | 212 ± 116 | 201 ± 108 |
| LWP (g m$^{-2}$) | 34 ± 17 (0.75) | 37 ± 43 (0.88) | 59 ± 54 (0.83) | 48 ± 47 (0.78) |
| King LWP (g m$^{-2}$) | 68 ± 30 (0.80) | 37 ± 35 (0.84) | 52 ± 40 (0.89) | 52 ± 38 (0.87) |
| LWP$_{ad}$ (g m$^{-2}$) | 77 ± 57 (0.97) | 51 ± 55 (0.96) | 93 ± 97 (0.94) | 79 ± 82 (0.93) |
| $C_w$ (g m$^{-3}$ km$^{-1}$) | 2.8 ± 0.3 | 2.1 ± 0.5 | 2.4 ± 0.4 | 2.7 ± 0.3 |
| $R_p$ (mm h$^{-1}$) | 0.02 ± 0.05 | 0.02 ± 0.08 | 0.10 ± 0.33 | 0.06 ± 0.25 |

Table 4: Average and standard deviation for cloud properties measured during contact and
separated profiles with 95 % confidence intervals (CIs) from a two-sample t-test applied to contact and separated profile data. Positive CIs indicate higher average for contact profiles and "insignificant" indicates statistically similar averages for contact and separated profiles.

| Parameter | Contact | Separated | 95 % CIs |
|---|---|---|---|
| $N_c$ (cm$^{-3}$) | 200 ± 103 | 113 ± 63 | 84 to 90 |
| $R_e$ (µm) | 7.5 ± 2.1 | 9 ± 3 | -1.6 to -1.4 |
| $\tau$ | 8.8 ± 8.3 | 7 ± 5 | 0.04 to 3.06 |
| LWC (g m$^{-3}$) | 0.23 ± 0.17 | 0.21 ± 0.14 | 0.01 to 0.02 |
| CWC (g m$^{-3}$) | 0.22 ± 0.16 | 0.20 ± 0.14 | 0.01 to 0.02 |
| RWC (x 10$^{-3}$ g m$^{-3}$) | 11 ± 15 | 18 ± 31 | -8 to -6 |
| H (m) | 194 ± 109 | 208 ± 106 | insignificant |
| LWP (g m$^{-2}$) | 46 ± 49 | 46 ± 41 | insignificant |
| CWP (g m$^{-2}$) | 45 ± 50 | 46 ± 44 | Insignificant |
| RWP (g m$^{-2}$) | 1.8 ± 3.3 | 3.0 ± 7.1 | -2.4 to -0.01 |
| $Z_T$ (m) | 1069 ± 267 | 1004 ± 271 | 6 to 123 |
| $Z_B$ (m) | 874 ± 294 | 796 ± 274 | 16 to 140 |
| $R_p$ (mm h$^{-1}$) | 0.04 ± 0.09 | 0.08 ± 0.33 | -0.05 to -0.03 |
| $S_{AUTO}$ (x 10$^{-10}$ s$^{-1}$) | 1.6 ± 3.0 | 4.9 ± 12.6 | -3.6 to -3.1 |
| $S_{ACC}$ (x 10$^{-8}$ s$^{-1}$) | 0.8 ± 1.6 | 1.7 ± 4.3 | -1.1 to -0.8 |
| $S_{ACC}/S_{AUTO}$ (x 10$^2$) | 0.7 ± 1.1 | 0.5 ± 0.9 | 0.2 to 0.3 |






Table 5: 95 % CIs from statistical comparisons between cloud regimes defined in text.

| Parameter | C-H relative to S-H | C-L relative to S-L |
|---|---|---|
| Above-cloud $N_a$ (cm$^{-3}$) | 852 to 948 | 387 to 413 |
| Below-cloud $N_a$ (cm$^{-3}$) | 194 to 226 | 45 to 53 |
| $N_c$ (cm$^{-3}$) | 98 to 108 | 25 to 31 |
| $R_e$ (μm) | -1.6 to -1.8 | -0.2 to -0.5 |
| $R_p$ (mm h$^{-1}$) | -0.03 to -0.04 | 0 to -0.04 |

Table 6: $S_o$ ± standard error for contact, separated, and all profiles, with sample size and $R$ in parentheses. $S_o$ is statistically insignificant if underlined.

| H | Contact | Separated | All Profiles |
|---|---|---|---|
| All | 0.87 ± 0.04 (173, 0.30) | 1.08 ± 0.04 (156, 0.36) | 0.88 ± 0.03 (329, 0.33) |
| 28 to 129 m | -0.06 ± 0.11 (52, -0.03) | 1.47 ± 0.10 (30, 0.55) | 0.67 ± 0.07 (82, 0.28) |
| 129 to 175 m | 0.88 ± 0.06 (38, 0.42) | 0.53 ± 0.09 (42, 0.20) | 0.68 ± 0.05 (80, 0.32) |
| 175 to 256 m | 0.92 ± 0.08 (41, 0.27) | 0.34 ± 0.07 (44, 0.13) | 0.54 ± 0.05 (85, 0.20) |
| 256 to 700 m | 1.15 ± 0.06 (42, 0.36) | 1.45 ± 0.07 (40, 0.41) | 1.13 ± 0.04 (82, 0.40) |


Table 7: $S_o$ ± standard error with sample size and $R$ in parenthesis for cloud regimes defined in text. $S_o$ is statistically insignificant if underlined.

| H | S-L | S-H | C-L | C-H |
|---|---|---|---|---|
| All | 1.29 ± 0.06 (107, 0.40) | 0.50 ± 0.06 (41, 0.19) | 0.86 ± 0.07 (40, 0.30) | 0.33 ± 0.05 (131, 0.11) |
| 28 to 129 m | 1.12 ± 0.15 (21, 0.42) | 0.43 ± 0.14 (8, 0.27) | 0.04 ± 0.42 (4, 0.01) | -0.33 ± 0.11 (48, -0.14) |
| 129 to 175 m | 0.66 ± 0.12 (25, 0.25) | 0.48 ± 0.18 (11, 0.17) | 0.50 ± 0.12 (9, 0.25) | 0.26 ± 0.08 (27, 0.13) |
| 175 to 256 m | 0.66 ± 0.09 (34, 0.22) | 0.07 ± 0.10 (9, 0.03) | 1.06 ± 0.13 (14, 0.34) | 0.61 ± 0.11 (27, 0.17) |
| 256 to 700 m | 1.89 ± 0.09 (27, 0.52) | 0.45 ± 0.11 (13, 0.14) | 0.72 ± 0.11 (13, 0.24) | 0.59 ± 0.09 (29, 0.17) |

Table 8: Meteorological and cloud properties from ERA5 reanalysis for contact, separated, and
all profiles with $LCC > 0.95$ (LCC is reported for all profiles), 95 % CIs from a two-sample t-test
applied to contact and separated profile data, and $R$ between each parameter and $LWP$ ($R_{LWP}$)
or $H$ ($R_H$) with statistically significant $R_H$ and $R_{LWP}$ in bold.

| Parameter | Contact | Separated | All | 95 % CIs | $R_H$, $R_{LWP}$ |
|---|---|---|---|---|---|
| LCC | 0.75 ± 0.29 | 0.83 ± 0.26 | 0.79 ± 0.28 | -0.14 to -0.02 | **0.24**, 0.04 |
| SST (K) | 293 ± 2 | 294 ± 3 | 293 ± 2 | -1.5 to -0 | **0.16, 0.22** |
| $H_{BL}$ (m) | 566 ± 164 | 624 ± 124 | 600 ± 144 | -103 to -11 | -0.05, -0.11 |
| ERA5 LWP (g m$^{-2}$) | 53 ± 18 | 51 ± 23 | 52 ± 21 | insignificant | **0.31, 0.18** |
| ERA5 RWP (g m$^{-2}$) | 0.71 ± 1.56 | 0.32 ± 0.40 | 0.48 ± 1.07 | 0.05 to 0.73 | **0.19**, -0.01 |
| $P_o$ (mb) | 1015 ± 1 | 1014 ± 2 | 1014 ± 2 | 1 to 2 | -0.09, -0.07 |
| $T_o$ (K) | 293 ± 2 | 293 ± 3 | 293 ± 2 | insignificant | **0.16, 0.27** |
| LTS (K) | 23 ± 2 | 22 ± 3 | 23 ± 3 | insignificant | -0.10, **-0.29** |
| EIS (K) | 8.1 ± 1.9 | 7.8 ± 3.1 | 7.9 ± 2.7 | insignificant | -0.13, **-0.31** |





Table A1: $S_o$ ± standard error with sample size and $R$ in parentheses for contact, separated, and
all profiles classified into a different number of populations.

| H Bin | Contact | Separated | All Profiles |
|---|---|---|---|
| **2 populations** | | | |
| 28 to 175 m | 0.53 ± 0.05 (90, 0.24) | 1.05 ± 0.07 (72, 0.39) | 0.69 ± 0.04 (162, 0.30) |
| 175 to 700 m | 1.06 ± 0.05 (83, 0.33) | 1.02 ± 0.05 (84, 0.33) | 0.93 ± 0.03 (167, 0.33) |
| **3 populations** | | | |
| 28 to 145 m | 0.08 ± 0.08 (67, 0.04) | 1.15 ± 0.09 (41, 0.45) | 0.60 ± 0.05 (108, 0.26) |
| 145 to 224 m | 0.95 ± 0.07 (51, 0.34) | 0.25 ± 0.06 (60, 0.11) | 0.60 ± 0.04 (111, 0.25) |
| 224 to 700 m | 1.08 ± 0.05 (55, 0.34) | 1.45 ± 0.06 (55, 0.41) | 1.05 ± 0.04 (110, 0.37) |

Table B1: $S_o$ ± standard error with sample size and $R$ in parentheses for contact, separated, and
all profiles with $R_p$ above a certain threshold.

| H Bin | Contact | Separated | All Profiles |
|---|---|---|---|
| **$R_p > 10^{-3}$ mm h$^{-1}$** | | | |
| All | 0.88 ± 0.03 (173, 0.34) | 0.95 ± 0.04 (156, 0.36) | 0.84 ± 0.02 (329, 0.37) |
| 28 to 129 m | 0.03 ± 0.10 (52, 0.02) | 1.41 ± 0.09 (30, 0.61) | 0.71 ± 0.07 (82, 0.33) |
| 129 to 175 m | 0.94 ± 0.05 (38, 0.49) | 0.64 ± 0.09 (42, 0.27) | 0.78 ± 0.04 (80, 0.40) |
| 175 to 256 m | 0.78 ± 0.07 (41, 0.30) | 0.21 ± 0.06 (44, 0.10) | 0.38 ± 0.04 (85, 0.18) |
| 256 to 700 m | 1.11 ± 0.06 (42, 0.38) | 1.18 ± 0.07 (40, 0.39) | 1.06 ± 0.04 (82, 0.42) |
| **$R_p > 10^{-2}$ mm h$^{-1}$** | | | |
| All | 0.49 ± 0.03 (173, 0.27) | 0.76 ± 0.03 (156, 0.38) | 0.61 ± 0.02 (329, 0.35) |
| 28 to 129 m | 0.01 ± 0.08 (52, 0.01) | 0.97 ± 0.10 (30, 0.57) | 0.48 ± 0.06 (82, 0.36) |
| 129 to 175 m | 0.70 ± 0.04 (38, 0.53) | 0.53 ± 0.08 (42, 0.29) | 0.66 ± 0.04 (80, 0.44) |
| 175 to 256 m | 0.62 ± 0.06 (41, 0.31) | 0.48 ± 0.05 (44, 0.31) | 0.47 ± 0.04 (85, 0.28) |
| 256 to 700 m | 0.37 ± 0.05 (42, 0.19) | 0.78 ± 0.06 (40, 0.33) | 0.60 ± 0.03 (82, 0.32) |

Table C1: $S_o$ ± standard error with sample size and $R$ in parenthesis for regimes defined in text
and different thresholds to define a low $N_a$ boundary layer. $S_o$ is statistically insignificant if
underlined.

| H | S-L | S-H | C-L | C-H |
|---|---|---|---|---|
| **Low $N_a$ = 300 cm$^{-3}$** | | | | |
| All | 1.37 ± 0.06 (96, 0.42) | 0.45 ± 0.06 (52, 0.17) | 0.29 ± 0.10 (21, 0.10) | 0.84 ± 0.04 (150, 0.29) |
| 28 to 129 m | 1.20 ± 0.16 (19, 0.44) | 0.38 ± 0.13 (10, 0.25) | NaN (0, NaN) | -0.06 ± 0.11 (52, -0.03) |
| 129 to 175 m | 0.68 ± 0.13 (21, 0.26) | 0.56 ± 0.16 (15, 0.20) | 0.02 ± 0.15 (6, 0.01) | 0.86 ± 0.07 (30, 0.41) |
| 175 to 256 m | 0.70 ± 0.10 (31, 0.24) | 0.07 ± 0.10 (12, 0.03) | 0.44 ± 0.17 (9, 0.15) | 1.04 ± 0.10 (32, 0.30) |
| 256 to 700 m | 2.03 ± 0.10 (25, 0.55) | 0.40 ± 0.10 (15, 0.12) | -0.09 ± 0.17 (6, -0.03) | 1.13 ± 0.07 (36, 0.36) |
| **Low $N_a$ = 400 cm$^{-3}$** | | | | |
| All | 1.12 ± 0.05 (125, 0.36) | 0.37 ± 0.09 (23, 0.16) | 1.11 ± 0.05 (64, 0.39) | 0.25 ± 0.06 (107, 0.08) |
| 28 to 129 m | 1.04 ± 0.13 (23, 0.43) | -0.20 ± 0.21 (6, -0.11) | 0.51 ± 0.22 (11, 0.21) | -0.33 ± 0.13 (41, -0.14) |
| 129 to 175 m | 0.81 ± 0.11 (30, 0.30) | 0.02 ± 0.19 (6, 0.01) | 0.90 ± 0.10 (12, 0.43) | 0.22 ± 0.09 (24, 0.10) |
| 175 to 256 m | 0.53 ± 0.09 (35, 0.19) | 0.12 ± 0.12 (8, 0.06) | 0.84 ± 0.09 (24, 0.30) | 0.53 ± 0.19 (17, 0.12) |
| 256 to 700 m | 1.42 ± 0.07 (37, 0.41) | 1.10 ± 0.42 (3, 0.25) | 1.52 ± 0.08 (17, 0.50) | 0.47 ± 0.09 (25, 0.13) |





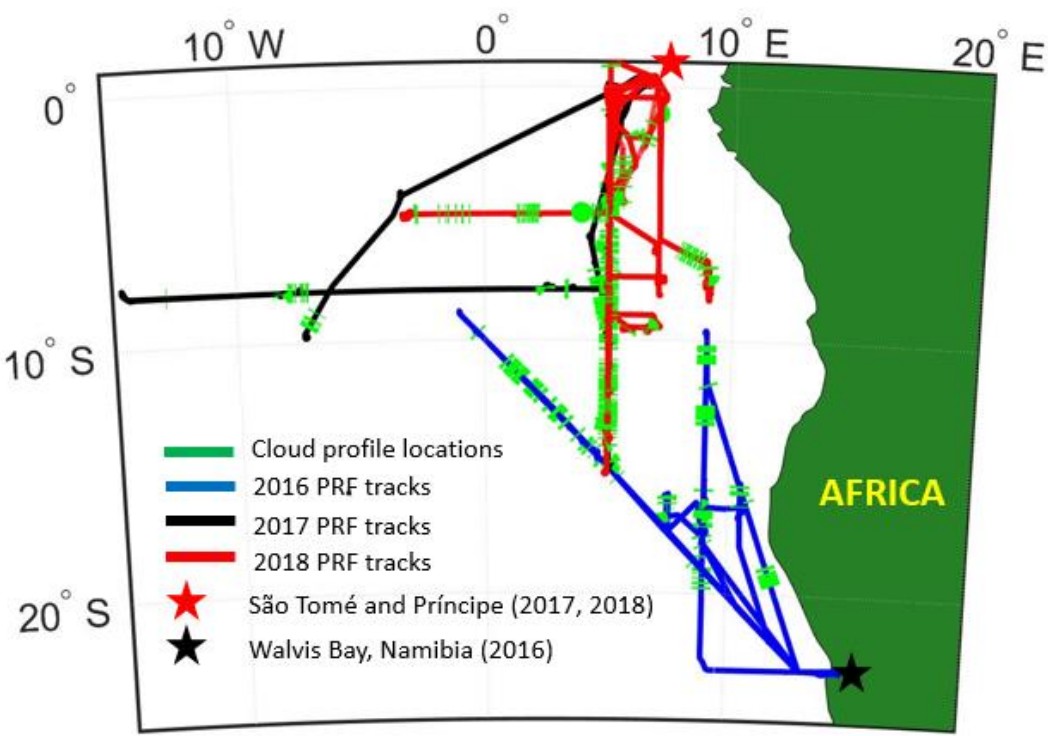

Figure 1: PRF tracks from ORACLES IOPs with base of operations and cloud sampling locations (tracks for multiple 2017 and 2018 PRFs overlap along 5° E).



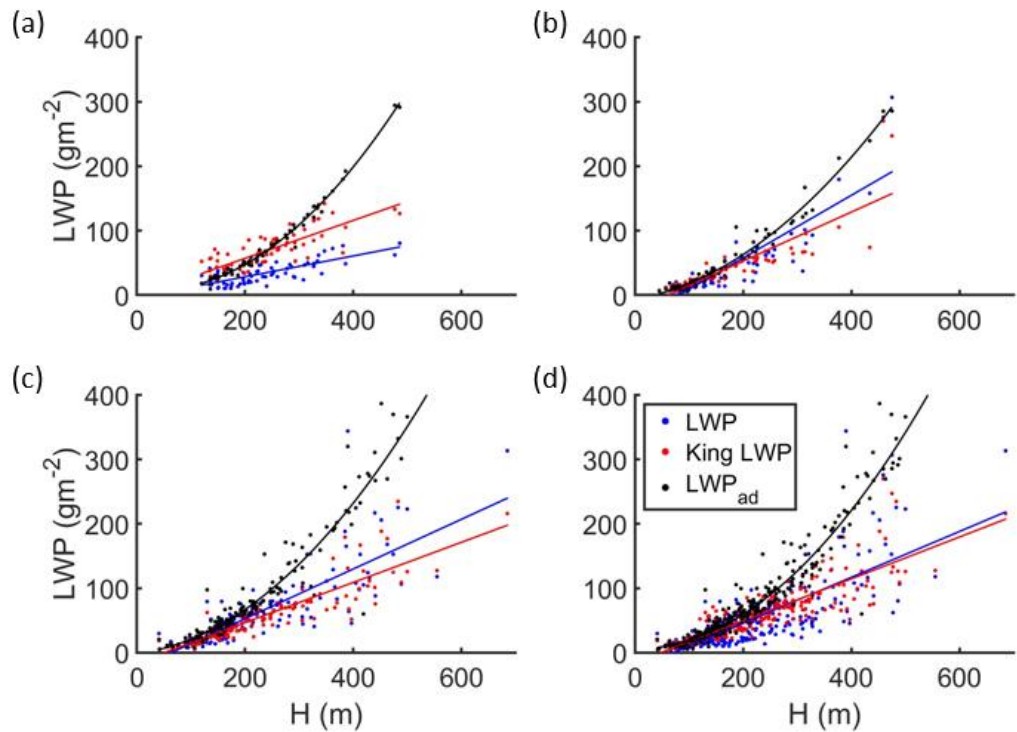

Figure 2: LWP from size-resolved probes, King LWP from the hot-wire, and adiabatic LWP (LWP$_{ad}$) for profiles with $LWP_{ad}$ > 5 g m$^{-2}$ as a function of $H$ for (a) 2016, (b) 2017, (c) 2018, and (d) all years with best-fit curves from a regression model applied to each LWP versus $H$.



Figure 3: Kernel density estimates (indicated by the width of shaded area) and boxplots showing the 25th, 50th (white circle), and 75th percentiles for (a) $N_c$, (b) $R_e$, (c) CWC, and (d) RWC as a function of $Z_N$ for contact and separated profiles.



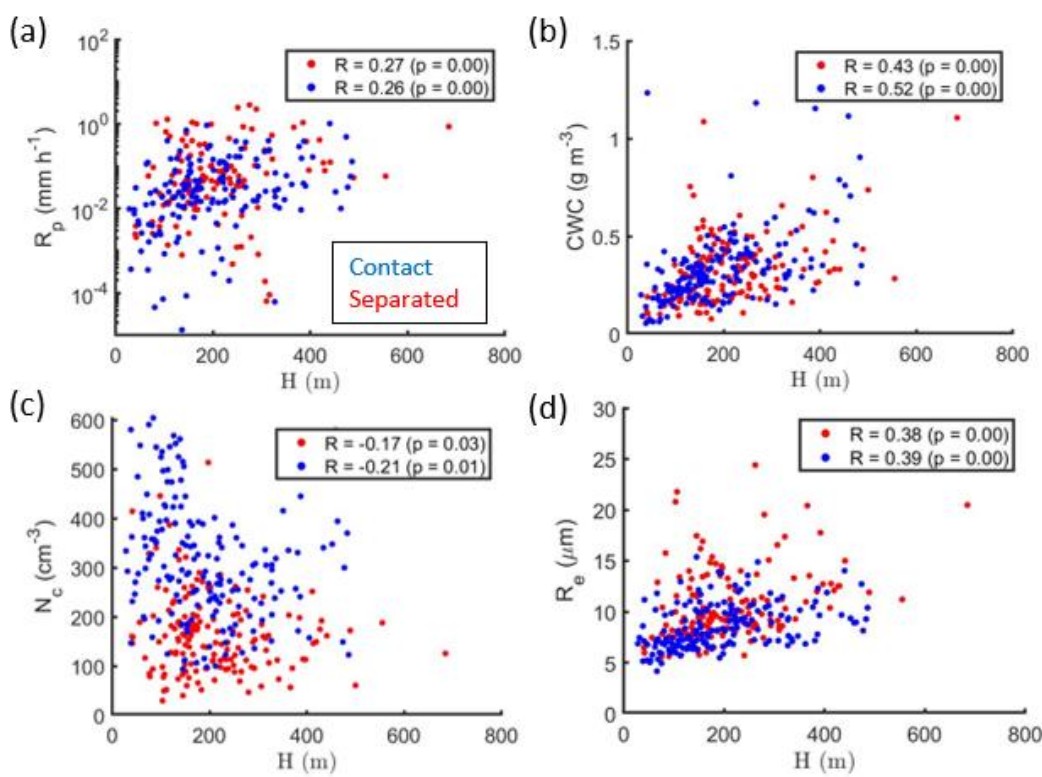


Figure 4: The 95th percentile for (a) $R_p$, (b) CWC, (c) $N_c$, and (d) $R_e$ as a function of $H$. Each dot represents the 95th percentile from the 1 Hz measurements for a single cloud profile. Pearson's correlation coefficient ($R$) and p-value for the correlation indicated in legend.

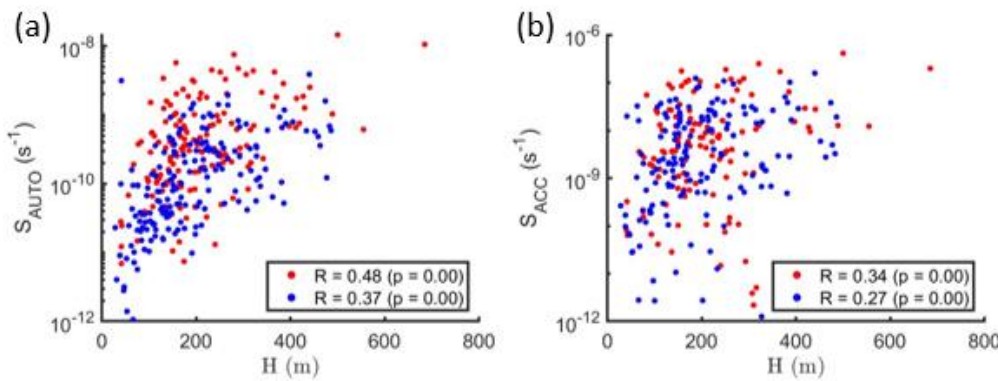


Figure 5: The 95th percentile for (a) $S_{AUTO}$ and (b) $S_{ACC}$ as a function of $H$. Each dot represents the 95th percentile from the 1 Hz measurements for a single cloud profile. $R$ and p-value for the correlation indicated in legend.



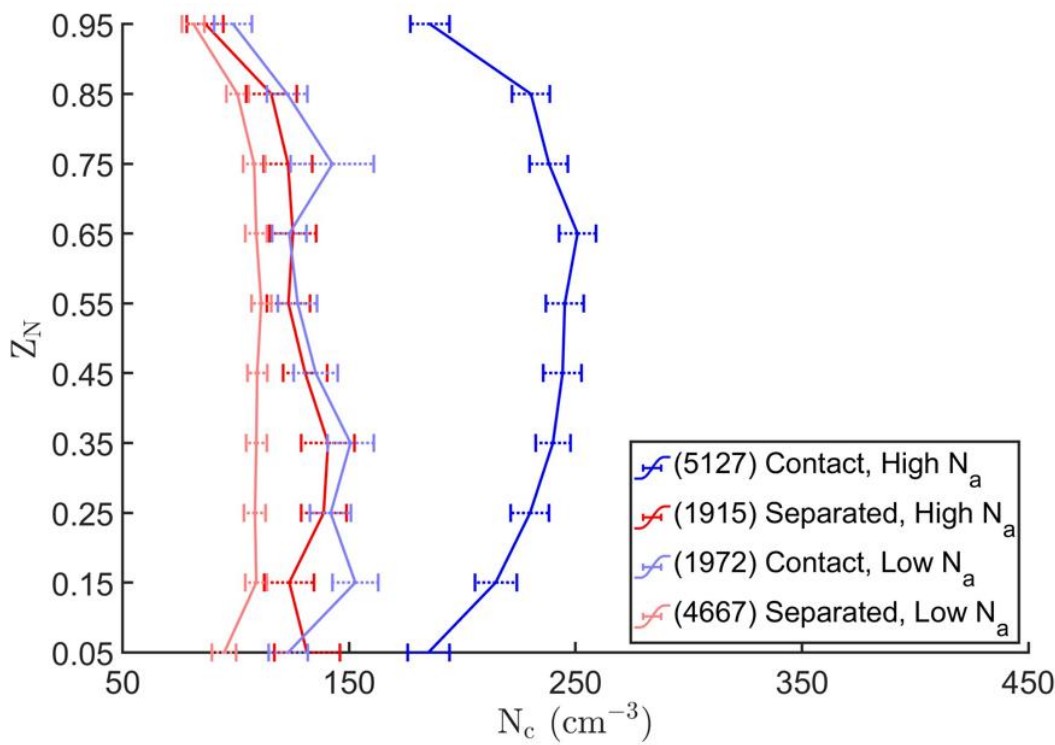

Figure 6: Average $N_c$ (error bars extend to 95 % CIs) as a function of $Z_N$. Number of 1 Hz data points and corresponding regimes indicated in legend.

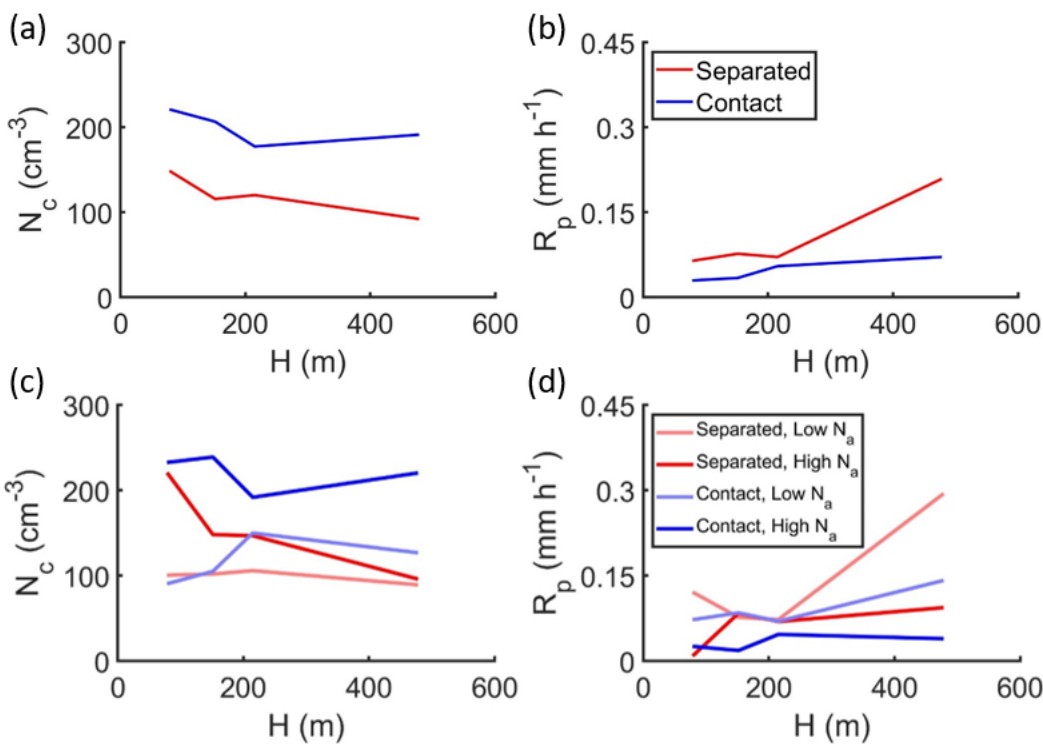

Figure 7: The average (a, c) $N_c$ and (b, d) $R_p$ as a function of $H$ for (a, b) contact and separated profiles, and (c, d) the regimes indicated in legend.






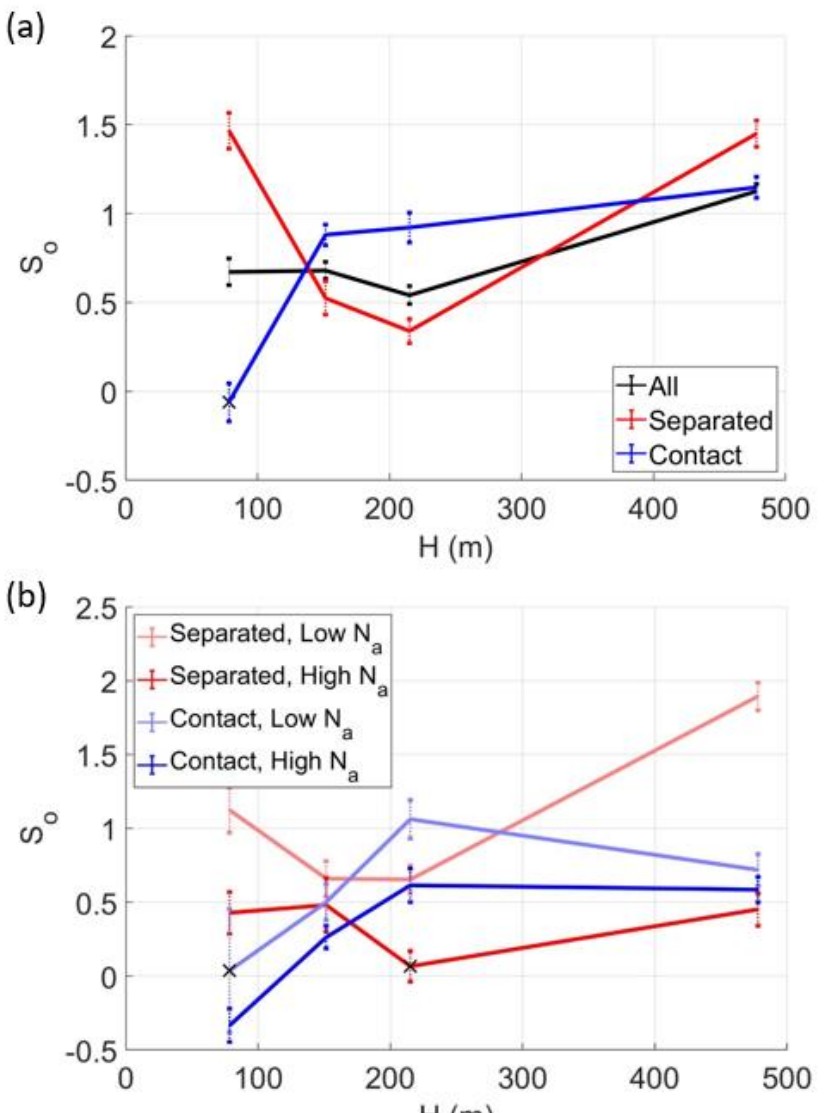

Figure 8: $S_o$ as a function of $H$ (error bars extend to standard error from the regression model) for (a) contact, separated, and all profiles, and (b) the regimes indicated in legend. $S_o$ was statistically insignificant when marked with a cross.





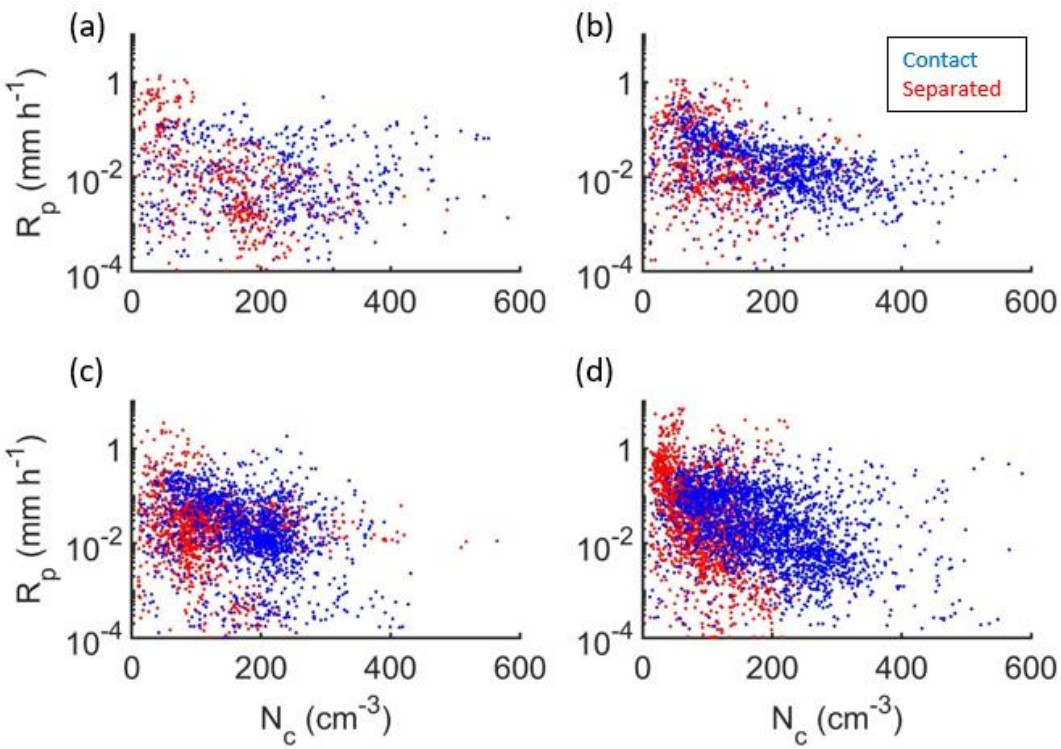

Figure 9: Scatter plots of $R_p$ and $N_c$ for 1 Hz data points from contact and separated profiles with (a) 28 < $H$ < 129 m, (b) 129 < $H$ < 175 m, (c) 175 < $H$ < 256 m, and (d) 256 < $H$ < 700 m.






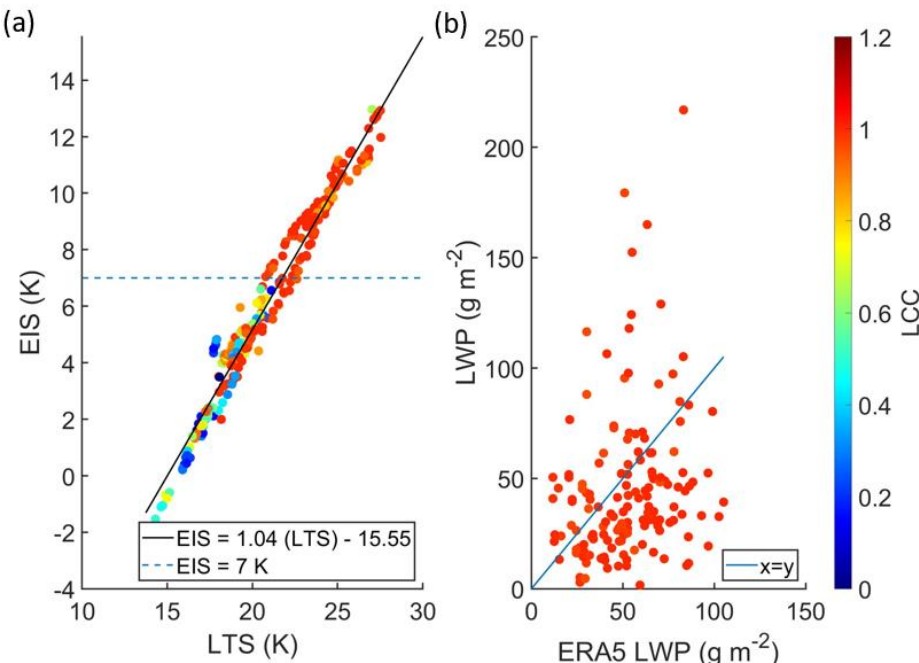

Figure 10: (a) LTS versus EIS with regression coefficients in legend ($R$ = 0.98) and (b) LWP from
size-resolved probes versus LWP from the ERA5 reanalysis ($R$ = 0.18) where each dot represents
a single cloud profile. LTS, EIS, ERA5 LWP, and LCC for each cloud profile taken from the nearest
ERA5 grid box (within 0.25° of latitude and longitude) at 12:00 UTC. Panel (a) shows all cloud
profiles and panel (b) shows cloud profiles with LCC > 0.95.



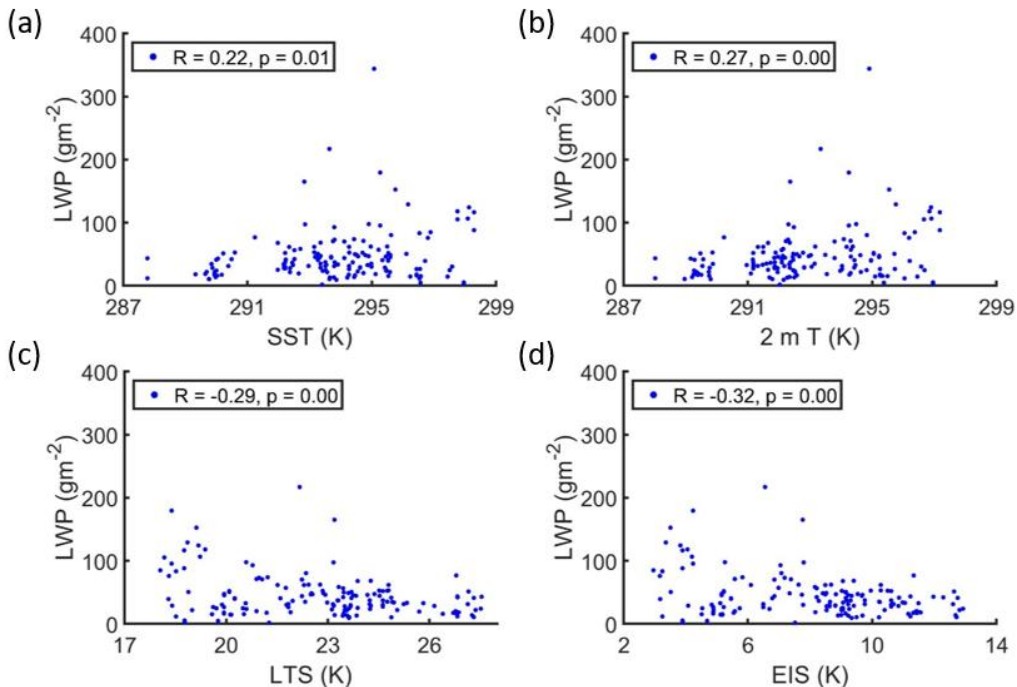

Figure 11: LWP from size-resolved probes as a function of (a) SST, (b) 2 m *T*, (c) LTS, and (d) EIS. Each dot represents a single cloud profile with LCC > 0.95 and SST, 2 m *T*, LTS, and EIS taken from the nearest ERA5 grid box (within 0.25° of latitude and longitude) at 12:00 UTC.





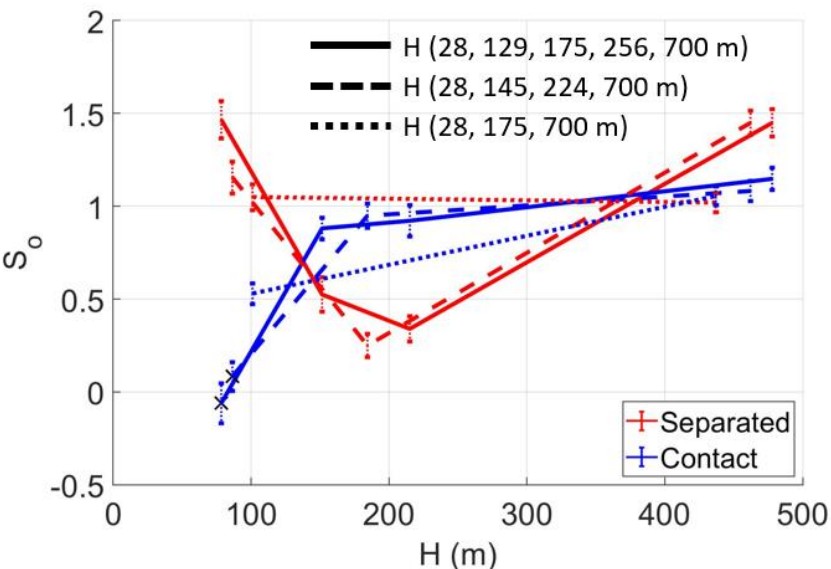

Figure A1: $S_o$ as a function of $H$ for contact and separated profiles classified into different populations using the end points indicated in legend. $S_o$ was statistically insignificant when marked with a cross.

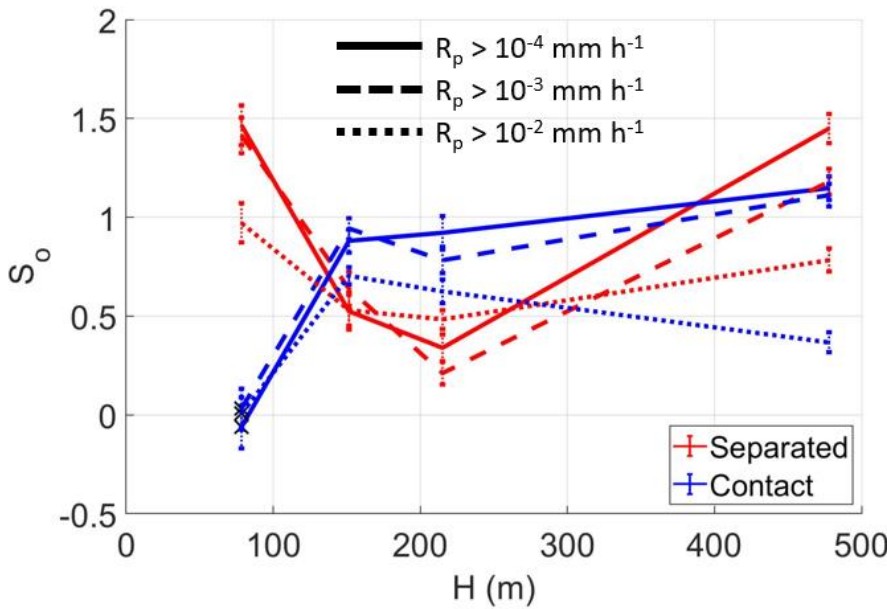

Figure B1: $S_o$ as a function of $H$ for contact and separated profiles with $R_p$ greater than the
thresholds indicated in legend. $S_o$ was statistically insignificant when marked with a cross.


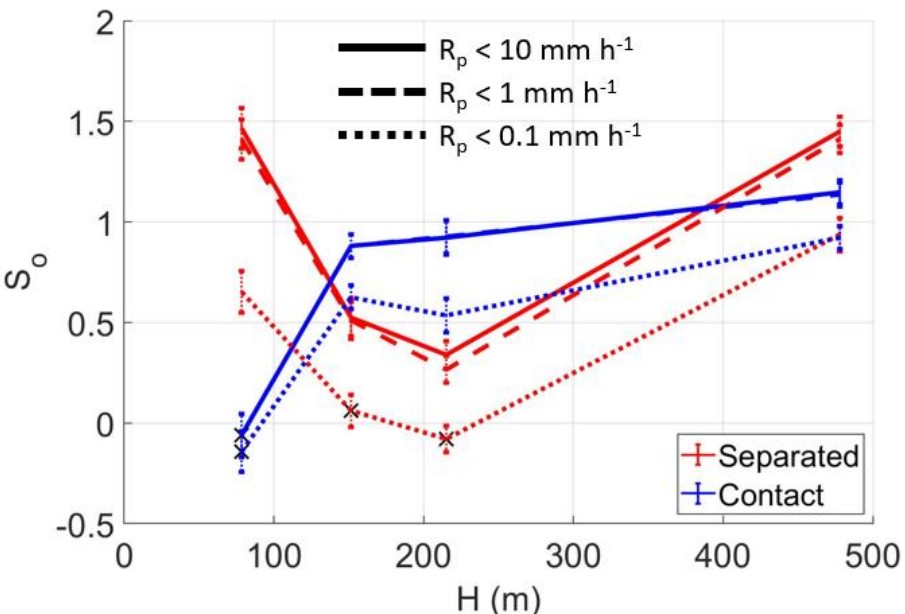

Figure B2: $S_o$ as a function of $H$ for contact and separated profiles with $R_p$ less than the thresholds indicated in legend. $S_o$ was statistically insignificant when marked with a cross.








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
