# Peer review of "Factors Affecting Precipitation Formation and Precipitation Susceptibility of Marine Stratocumulus with Variable Above and Below-Cloud Aerosol Concentrations over the Southeast Atlantic"

_Atmospheric Chemistry and Physics, 2021_

## Author Comment (AC1)

This document includes the reviewer comments (red) and author responses (black).

Review of "Precipitation Susceptibility of Marine Stratocumulus with Variable Above and Below-Cloud Aerosol Concentrations over the Southeast Atlantic" by Gupta et al.

This paper presents airborne observations from the ORACLES project that examine how cloud and precipitation characteristics vary with perturbations driven largely from the entrainment of free-tropospheric biomass burning aerosols into the southeast Atlantic marine boundary layer. The authors extend their previous study by incorporating a much larger observational dataset from additional flight years and they extend their prior work to look at precipitation susceptibility. I commend the authors for synthesising such a large dataset and I found the paper to be generally well written. The topic area is certainly suitable for publication in ACP. However, I do have a major concern about the use of the CAS probe to measure liquid water content and cloud drop size from the 2016 campaign (see below), that I feel the authors need to address before this paper can be published.

The authors thank the reviewer for their thorough review. These suggestions will improve the quality of the submitted manuscript. Each of the reviewer's comments has been addressed in this document.

Main concern

1. Use of CAS probe for 2016 flights. I have concerns about the use of the CAS data for calculating microphysical properties on the 2016 campaign. Although the cloud drop number concentration from the CAS looks reasonable when compared against the PDI (Fig S1a), the LWC looks to be underestimated when compared against both the PDI and King probes (Fig S1b, Fig S2). The authors show that the PDI can give higher LWC values when compared to the adiabatic value and so choose not to use that instrument. But given that the bulk LWC estimate from the King probe is also much higher that the CAS (Fig S2), I think the authors need to provide some additional justification as to why the CAS probe is thought to be reliable (for measurements of LWC and effective radius). A possible approach would be to look at cases where the cloud was expected to be more adiabatic (well mixed boundary layer, non-drizzling etc) and examine if the difference with the adiabatic LWC value shown in Fig S2 is still apparent. Alternatively, in precipitating clouds, can the overlap with the 2DS probe be looked at to at least check for consistency at the larger cloud drop sizes that contribute significantly to LWC. I also note however that a similar low bias in LWC from the CAS is shown in the 2017 and 2018 campaigns when compared against a CDP (Fig S4 and S6), which suggests that it could be a general measurement issue with the CAS measurements. Related to this point, if the authors removed the 2016 data from their analysis, do any of the conclusions of the paper change?

The main concern regarding the use of CAS data for the 2016 campaign is justified given the differences between the CAS and PDI datasets for 2016 and the CAS and CDP datasets for 2017/2018. Specific portions of the main concern are addressed:

40 A possible approach would be to look at cases where the cloud was expected to be more adiabatic (well mixed boundary layer, non-drizzling etc) and examine if the difference with the adiabatic LWC value shown in Fig S2 is still apparent.

Contact profiles underwent precipitation suppression and were not drizzling as heavily as separated profiles. However, the differences between CAS LWC and the adiabatic LWC were observed during both separated and contact profiles. It is unlikely the differences between CAS LWC and the adiabatic LWC were dependent on the adiabaticity of the clouds.

in precipitating clouds, can the overlap with the 2DS probe be looked at to at least check for consistency at the larger cloud drop sizes that contribute significantly to LWC

During frequent flight legs through regions with high aerosol concentration ($N_a$), soot deposition occurred on the optical lenses of the 2D-S probe which led to data artifacts. These artifacts were removed using additional constraints during data processing (Gupta et al., 2021). Due to these constraints, 2D-S measurements for droplets with diameter ($D$) below 50 μm were unusable. Since the CAS measurement range does not exceed 50 μm, the CAS and 2D-S datasets were not compared. Comparisons between the closest CAS and 2D-S size bins are shown below for cloud profiles from 6 September 2016 (within the response to minor comment number 13).

if the authors removed the 2016 data from their analysis, do any of the conclusions of the paper change?

If 2016 data were removed, there were minor changes in the trends of $S_o$ versus cloud thickness ($H$) and negligible changes in the $S_o$ differences between contact and separated profiles (Fig. 1). Most noticeably, there was a decrease in $S_o$ for thick clouds ($H > 256$ m) with a negligible decrease in the overall $S_o$ (Table 1). This was accompanied by a decrease in the number of thick clouds when the 2016 data were removed. These changes highlight that the discussion of $S_o$ in the study was robust as it relates to the exclusion of the 2016 data. In the absence of CDP data, the $N_c$ and LWC from the CAS and PDI datasets were compared to choose which dataset should be used to represent the 2016 campaign. These comparisons were quantified in the supplement for transparency.

$S_o$ was calculated using PDI data from the 2016 campaign to determine the impact of choosing CAS data over PDI data. When the PDI data were used, there were minor changes in the trends of $S_o$ versus $H$ (Fig. 1) and the magnitude of $S_o$ for different aerosol regimes (Table 1). This was because $S_o$ depends on $N_c$ and precipitation rate ($R_p$). The CAS and PDI datasets had small differences in the average $N_c$ over the 2016 campaign (95 % confidence intervals of 9 to 12 cm$^{-3}$) and $R_p$ was calculated using droplets with $D > 50$ μm which did not include contributions from either the CAS or the PDI. Since the 2016 campaign contributed about a third of the ORACLES

measurements, data from the 2016 campaign were included in the study so as not to reduce the size of the dataset.

I also note however that a similar low bias in LWC from the CAS is shown in the 2017 and 2018 campaigns when compared against a CDP (Fig S4 and S6), which suggests that it could be a general measurement issue with the CAS measurements.

There could be a sizing bias in the CAS given the small differences in $N_c$ and large differences in *LWC* from the CAS and PDI datasets (see supplement). The sizing bias could impact the quantitative results presented in the study. Since $R_p$ does not depend on CAS LWC, the impact of the sizing bias would be limited to effective radius ($R_e$) and LWC for droplets with $D < 50$ μm (cloud water content or CWC). Assuming the King hot-wire provided an accurate estimate of LWC for the sampled clouds, the CAS droplet size distribution could be adjusted using the King LWC following the methodology by Painemal and Zuidema (2011). For the 2016 research flights, the King LWC and the CAS LWC had a best fit slope ($a$) between 0.46 and 0.63. The CAS *n(D)* was scaled by adjusting the CAS size bins as

$$CAS\ LWC = a\ x\ King\ LWC,\ \ D_i^* = a^{-1/3}\ D_i\ ,\ \ \ \ \ \ \ \ \ \ \ \ \ \ \ \ \ \ \ \ \ \ (AC1)$$

where $D_i$ is the bin midpoint for the $i^{th}$ bin and $D_i*$ is the scaled bin midpoint. CAS bin midpoints increased by up to 30 % since $R_i^* > R_i$ for $a < 1$ and each flight had $a < 1$. This led to higher $R_e$ and CWC for both contact and separated profiles (Fig. 2). The difference between the average $R_e$ for contact and separated profiles increased from 1.5 μm to 1.7 μm when CAS data were scaled (Table 2). The difference in the average LWC decreased from 0.02 g m$^{-3}$ to 0 when CAS data were scaled. Scaling the CAS data thus would not have a large impact on the results.

To avoid confusion with the use of two datasets from the CAS (original or scaled using King LWC), the authors used the original CAS dataset in the study. Given the minor impact of using the PDI/CAS/no data on $S_o$ (Table 1), the original CAS dataset could be used. The differences between the datasets from different probes were quantified in the supplement. We believe these responses addressed any concern regarding the use of CAS data from the 2016 campaign. We believe the documentation of the differences between the ORACLES cloud probes in the supplement provides appropriate information about the uncertainties associated with any analyses using the ORACLES in situ cloud probes.

Minor comments

1.  Line 34: Suggest changing to "changes in microphysical properties" in this sentence as the proceeding sentence states that LWP and cloud thickness are similar. Also it is not clear what the reference to "existing relationships" means. Which relations are you referring to? Are these based on previous observations or parameterized/simulated in models for example?

The sentence was updated by adding the underlined words: "These results suggest the changes in cloud microphysical properties were driven by ACIs rather than meteorological

effects, and the existing relationships between $R_p$ and $N_c$ in model parameterizations must be adjusted to account for the role of ACIs".

2. Line 68: assuming constant LWP in what?

The sentence was updated by adding the underlined words: "Since $\tau$ has greater sensitivity to LWP compared to $N_c$, assuming constant LWP under different aerosol conditions can lead to underestimation of the cloud albedo susceptibility to aerosol perturbations (Platnick and Twomey, 1994; McComiskey and Feingold, 2012)."

3. Line 78: sink of liquid water rather than LWP?

"LWP" was changed to "LWC".

4. Line 82: Suggest changing to "relates the change in Rp…." as the actual definition is Eq 9.

The sentence was updated.

5. Line 87: Suggest changing to "parameterized in models"

The sentence was updated.

6. Line 93: Suggest changing to something like "A focus of recent field experiments in the southeast Atlantic Ocean has been to study ACI in this unique meteorological ……..".

The sentence was updated to: "Recent field campaigns focused on studying ACIs over the southeast Atlantic Ocean because unique meteorological conditions are present in the region"

7. Line 98: I assume the values of single scattering albedo and above cloud optical depth are from ORACLES. Please make that clear. Much higher optical depths can occur in the region e.g. Peers at al. (2021).

Lines 110-115 were moved ahead of the sentence and the sentence was updated to: "During ORACLES, the aerosol layer was comprised of shortwave-absorbing aerosols (500 nm single-scattering albedo of about 0.83) with above-cloud aerosol optical depth up to 0.42."

8. Line 102: Suggest changing "positive forcing" to "aerosol absorption of SW radiation". Also please expand on why this decreases entrainment e.g. strengthening inversion.

The sentence was updated to: "Warming aloft due to aerosol absorption of solar radiation strengthens the temperature inversion which decreases dry air entrainment into clouds, increases LWP and cloud albedo, and decreases the shortwave CRF".

9. Line 149: Please include references for the different probes.

140    The references were added.

The sentence was updated.

11. Line 165: What thresholds on the PCASP data were used to screen cloud?

This was moved after Line 209 where in-cloud measurements thresholds were defined.

145

We have left this unchanged given that CAS n(D) was used for ORACLES 2016 and CDP n(D) was used for ORACLES 2017 and 2018.

13. Line 170: Can the authors comment on how well the different probes compared for drop sizes where they overlap?

150    The probes generally had good agreement in the overlap regions. This was determined by comparing the number distribution function, $N(D)$, from different probes. Figure 3 shows the sawtooth flight patterns flown to sample clouds on 6 September 2016. Figure 4 shows the average $N(D)$ from the CAS, the 2D-S, and the HVPS-3 during these flight patterns. The largest difference between $N(D)$ from the largest CAS size bin and the smallest 2D-S size bin was sampled
155    during the 3$^{rd}$ sawtooth when the average CAS $N(D)$ was $3.1 \times 10^{-3}$ cm$^{-3}$ μm$^{-1}$ and the average 2D-S $N(D)$ was $6.2 \times 10^{-3}$ cm$^{-3}$ μm$^{-1}$. It is difficult to determine the differences between the 2D-S and the HVPS-3 because large droplets with sizes where these probes have good overlap (800 to 1200 μm) were very rarely sampled during the ORACLES research flights. For the 3$^{rd}$ sawtooth, the 2D-S $N(D)$ from the size bin centered near 450 μm was $22.9 \times 10^{-9}$ cm$^{-3}$ μm$^{-1}$ and the corresponding
160    HVPS-3 $N(D)$ was $8.8 \times 10^{-9}$ cm$^{-3}$ μm$^{-1}$.

The sentence was updated.

15. Line 215: Please briefly outline what you mean by shallow boundary layers can provide an underestimate of LWP adjustments. And I don't understand what you mean in the
sentence on line 217. Please clarify.

This text (Line 214-218) was removed for brevity.

16. Line 219: Have you done any analysis of the thermodynamic data to ascertain the frequency of well-mixed vs decoupled boundary layers from the vertical profiles used in this study? And are the clouds studied typically a single layer of stratocumulus, rather
than cumulus rising into stratocumulus. If the latter form a significant number of profiles,

can the authors comment on how that may impact the results e.g. cumulus could transport aerosol from the surface mixed layer up into the overlying cloud.

Such analysis was not conducted due to the cloud sampling strategy during ORACLES. A complete sounding of the boundary layer or the free troposphere was not conducted near most profiles. Instead, cloud sampling was conducted in a sawtooth pattern where the aircraft ascended or descended through the cloud layer in quick succession. During the sawtooth patterns, the aircraft only flew up to about 100 m above or below cloud to maximize cloud sampling within a short distance (Fig. 3). Therefore, if a cumulus layer was present, it was not always sampled. When multiple cloud layers were sampled during a sawtooth pattern, a single cloud layer that capped the boundary layer was selected to limit data analysis to stratocumulus clouds. Thermodynamic analyses based on in situ measurements were also hindered by concerns about aircraft measurements of dew point and mixing ratio during descents (Gupta et al., 2021).

17. Equation 1: Is effective radius calculated as a function of height or is it a cloud top value.

Effective radius ($R_e$) was calculated for each 1 Hz data sample within cloud. Equation 1 was updated to avoid confusion:

$$\text{“}R_e\,(h) = \int_3^\infty D^3\,N(D,h)\,dD \Big/ \int_3^\infty 2\,D^2\,N(D,h)\,dD\ . \tag{1}\text{”}$$

The following text was added: "Based on the aircraft speed, 1 Hz data samples corresponded to roughly 5 m intervals in the vertical direction."

18. Equation 4: What does LWC(zB) mean? Is it your fixed value of 0.05 g m-3 to define cloud base?

This could be any value greater than 0.05 g m$^{-3}$. Equation 4 was removed for brevity.

19. Line 283/Table 4: Is the effective radius calculated at cloud top, or is it an average value through the depth of the cloud?

$R_e$ was calculated for each 1 Hz data sample within cloud. During a cloud profile, this meant a value for effective radius was calculated at 5 m intervals. These values were then averaged over the entire depth of the cloud.

20. Line 377: Suggest changing "ascent" to "updraft"

The sentence was updated by adding the underlined words: "Alternatively, there may not have been sufficient time for the updraft to produce the few large droplets needed to broaden the size distribution and initiate collision-coalescence.

21. Line 385: Suggest changing "explained" to "parameterized"

The sentence was updated.

22. Figure 5: y-axis labels should be SAUTO95 and SACC95

The labels were updated.

205     23. Line 420: Add comment as to why you use CO as a proxy for the airmass from biomass burning aerosol source regions.

The following text was added: "Carbon monoxide (CO) concentrations were examined since CO acts as a biomass burning tracer that is unaffected by precipitation scavenging (Pennypacker et al., 2020)."

210     24. Line 422: Suggest the authors may want to comment on how the different regimes occur. For example, does the S-H regime have high boundary layer aerosol loadings because of previous entrainment events prior to the aircraft sampling and does the C-L regime have low aerosol loadings because there hasn't been sufficient time for aerosol to be mixed down into the boundary layer from the overlying aerosol plume.

215     Pennypacker et al. (2020) found the boundary layer CO concentrations during the biomass burning season (July to October) were elevated compared to CO concentrations from the non-burning season (December to April). They argued the boundary layers were previously polluted by biomass-burning aerosols with low aerosol concentration ($N_a$) due to precipitation scavenging. The C-L regime could be explained by precipitation scavenging since these boundary
220     layers also had elevated CO concentrations. Aerosol-induced precipitation suppression would explain why the number of C-L cases was much less than C-H cases. The S-H regime could be hypothesized to be associated with prior entrainment events (Diamond et al., 2018). We have not added these statements to the text because of their speculative nature given that the studies referenced examined boundary layers at a different location (Ascension island) and flight days.

225     25. Line 433: Figure 6 does not show comparisons of Re as stated in the text.

The sentence was updated by adding a reference to Table 5 for the comparisons of $R_e$.

26. Line 443: The authors test boundary layer aerosol concentration thresholds of 300 to 400 cm-3 to split "low" and "high" aerosol conditions. Even the value of 300 cm-3 seems like a moderately polluted boundary layer though compared to pristine marine conditions,
230     where I would expect values < 100 cm-3 to be typical. Did ORACLES measure cleaner boundary layer conditions and if yes, how do the cloud properties in these cleaner clouds compare to the broader "low" aerosol regime used in this study? Or do future studies need to compare/contrast with more offshore airborne measurements from the CLARIFY campaign for example?

235     Ground-based observations at Ascension Island showed monthly accumulation mode $N_a$ around 200 cm$^{-3}$ in 2016 and around 400 cm$^{-3}$ in 2017 during the biomass burning season (Pennypacker et al., 2020). These values suggest our definition of low $N_a$ was reasonable since

these measurements were collected during the peak of the burning season. Only 2 out of the 329 profiles used in this study (having 40 1-Hz in-cloud samples) were flown within a boundary layer with $N_a < 100$ cm$^{-3}$. These profiles had average below-cloud $N_a = 53$ cm$^{-3}$ and average $N_c = 23$ cm$^{-3}$. Given the small number of data samples, measurements outside the burning season may be needed to compare the data used in this study with pristine boundary layers.

27. Line 448: Is quartiles the correct term, given that the bins don't have an equal number of profiles (82,80,85,82 in Table 6)?

The sentence was updated to: "The populations were divided at $H = 129$, 175, and 256 m to ensure similar sample sizes (Table )."

28. Figure 9: It looks like there are data points with Nc = 0 cm-3. Is that correct?

There were no data points with $N_c$ below 10 cm$^{-3}$ even though it may seem like they have values close to zero. The definition of in-cloud measurements ($N_c > 10$ cm$^{-3}$ and King LWC > 0.05 g m$^{-3}$) was used to screen each data sample used in the study.

29. Line 489: Suggest adding "decreased with H from …..include your numbers for the lowest H bin data….to 0.53"

The sentence was updated by adding the underlined text: "For separated profiles, $S_o$ decreased with $H$ from 1.47 ± 0.10 for $H < 129$ m to 0.53 ± 0.09 for 129 < $H$ < 175 m and to 0.34 ± 0.07 for 175 < $H$ < 256 m".

30. I found some of the text hard to follow in section 6.3, that was at least in part due to the number of figures that were referred to in short succession. For example, the short paragraph beginning on line 489 refers to four different figures. I would suggest the authors consider rewriting some of the text and highlighting key points, to make it easier for the reader.

The text in Section 6.3 has been edited for clarity based on this comment.

31. Line 491: I struggle to see how the reader can see this change by looking at Fig 9 b,c.

The reference to Figure 9 is removed.

32. Line 513: The paragraph starts by mentioning appendix B, but does not then summarize the key point of that sensitivity study. Further, it states that the appendix investigates the inclusion of precipitating clouds, but I think it instead looks at the impact of the removal of non-precipitating clouds.

The text was updated to: "The sensitivity of $S_o$ to removal of clouds based on $R_p$ was examined in Appendix B. The removal of clouds with low $R_p$ and high $N_c$ or with high $R_p$ and low $N_c$ resulted in lower average $S_o$ consistent with previous work (Duong et al., 2011)."

33. Line 600: What is the mechanism that results in higher RWP for these contact profiles?

This is because a subset of the contact and separated profiles was compared. A physical basis for increased RWP in polluted conditions for model low-cloud cover > 0.95 is not expected.

34. Line 625: Suggest rephrasing this statement. The separated polluted boundary layers have presumably also experienced entrainment events prior to the aircraft sampling, even though there is no contact at the time of the measurements. The timescales of entrainment and history of the airmass do also need to be considered (Diamond et al., 2018), rather than just an instantaneous measure of "contact" vs "separated".

Conclusion #2 was changed to: "Aerosol-induced cloud microphysical changes in both clean and polluted boundary layers." The following line was updated: "Contact profiles were more often located in polluted boundary layers and had higher below-cloud CO concentration (27 to 29 ppb higher) which suggests more frequent entrainment of biomass-burning aerosols into the boundary layer compared to separated profiles."

35. Line 630: Make it clear that this is when compared to separated profiles.

The sentence was updated by adding the underlined words: "Contact profiles had 25 to 31 cm$^{-3}$ higher $N_c$ and 0.2 to 0.5 μm lower $R_e$ in clean and 98 to 108 cm$^{-3}$ higher $N_c$ and 1.6 to 1.8 μm lower $R_e$ in polluted boundary layers compared to separated profiles."

References

Diamond, M. S., et al., 2018. Atmos. Chem. Phys., https://doi.org/10.5194/acp-18- 14623-2018, 2018.

Peers, F., et al., 2021. Atmos. Chem. Phys., 21, 3235–3254, https://doi.org/10.5194/acp-21-3235-2021.

REFERENCES:

Gupta, S., McFarquhar, G. M., O'Brien, J. R., Delene, D. J., Poellot, M. R., Dobracki, A., Podolske, J. R., Redemann, J., LeBlanc, S. E., Segal-Rozenhaimer, M., and Pistone, K.: Impact of the variability in vertical separation between biomass burning aerosols and marine stratocumulus on cloud microphysical properties over the Southeast Atlantic, Atmos. Chem. Phys., 21, 4615–4635, https://doi.org/10.5194/acp-21-4615-2021, 2021.

Painemal, D. and Zuidema, P.: Assessment of MODIS cloud effective radius and optical thickness
retrievals over the Southeast Pacific with VOCALS-REx in situ measurements, J. Geophys.
Res., 116, D24206, https://doi.org/10.1029/2011jd016155, 2011.

**TABLES AND FIGURES:**

Table 1: $S_o$ ± standard error for all profiles, with sample size and $R$ in parentheses.

| H | CAS data from 2016 | No data from 2016 | PDI data from 2016 |
|---|---|---|---|
| All | 0.88 ± 0.03 (329, 0.33) | 0.83 ± 0.03 (258, 0.33) | 0.90 ± 0.02 (329, 0.35) |
| 28 to 129 m | 0.67 ± 0.07 (82, 0.28) | 0.58 ± 0.07 (80, 0.26) | 0.68 ± 0.07 (84, 0.29) |
| 129 to 175 m | 0.68 ± 0.05 (80, 0.32) | 0.73 ± 0.05 (63, 0.35) | 0.73 ± 0.05 (79, 0.35) |
| 175 to 256 m | 0.54 ± 0.05 (85, 0.20) | 0.84 ± 0.06 (58, 0.31) | 0.71 ± 0.05 (86, 0.26) |
| 256 to 700 m | 1.13 ± 0.04 (82, 0.40) | 0.75 ± 0.04 (57, 0.30) | 1.10 ± 0.04 (80, 0.41) |

Table 2: Average and standard deviation for $R_e$ and CWC over all three ORACLES
deployments calculated with original CAS data from 2016 (underlined) and CAS data from 2016
scaled using King LWC (bold).

| Parameter | Contact | Separated |
|---|---|---|
| R_e (μm) | 7.5 ± 2.1 | 9.0 ± 3.0 |
| **R_e (μm)** | **7.8 ± 2.1** | **9.5 ± 2.9** |
| CWC (g m$^{-3}$) | 0.23 ± 0.17 | 0.21 ± 0.14 |
| **CWC (g m$^{-3}$)** | **0.24 ± 0.17** | **0.24 ± 0.15** |

[Figure]

310          Figure 1: $S_o$ as a function of $H$ (error bars extend to standard error from regression model) using (a) CAS data from 2016, (b) no data from 2016, and (c) PDI data from 2016.

[Figure]

Figure 2: Kernel density estimates (indicated by the width of shaded area) and boxplots showing the 25th, 50th (white circle), and 75th percentiles for $R_e$ and CWC over the three ORACLES deployments calculated using (a, b) original CAS data from 2016 and (c, d) CAS data from 2016 scaled using the King LWC.

[Figure]

Figure 3: P-3 aircraft altitude as a function of time during sawtooth flight patterns. Data are colored by accumulation mode aerosol concentration (taken from Gupta et al., 2021).

[Figure]

Figure 4: The average number distribution function *N(D)* from different probes for sawtooth patterns shown in Fig. 3.

---

## Author Comment (AC2)

This document includes the reviewer comments (red) and author responses (black).

- Does the paper address relevant scientific questions within the scope of ACP?

- The paper addresses "Precipitation Susceptibility of Marine Stratocumulus" to variations in aerosol concentrations. This is an important topic within the scope of ACP.

- Does the paper present novel concepts, ideas, tools, or data?

- The novel part of this research is the aircraft dataset. No new concepts, ideas or tools are introduced.

- Does the title clearly reflect the contents of the paper?

- The title places emphasis on precipitation susceptibility, which is only a small component of the paper, has questionable validity, and lacks proper discussion.

This comment is addressed by modifying the title to include "Factors affecting" at the beginning. The discussion of precipitation susceptibility ($S_o$) in Section 6.3 has been edited for clarity. The data analysis before quantifying $S_o$ in Section 6 supported the discussion of $S_o$ for different aerosol regimes. Figure 9 was added to make the link more explicitly with a summary in Section 6.4. The reviewer's concerns about methodology, data interpretation and validity of the results are addressed through this document.

- Does the abstract provide a concise and complete summary?

- The title emphasises "Precipitation Susceptibility". This is not mentioned at all in the abstract, but it is discussed in the conclusions.

The reviewer is directed to Lines 22-24 from the abstract. These lines mention $S_o$ and provide a quantitative estimate of the aerosol-induced change in $S_o$.

- Are substantial conclusions reached?

- The main results stem from figures 3 and 7, which demonstrate that there is some difference in the cloud microphysical properties between two subsets of the data. This is attributed to biomass burning aerosols entrained into the boundary layer from above. The paper also explores potential impacts of aerosols on precipitation formation, and the role of meteorological factors also affecting the microphysical properties. The conclusions are rather limited in scope, and in part unsubstantiated. I understand observational based research is important and also difficult to publish in its own right without complex modelling, so maybe a "Measurement Report" is more suitable format?

Modelling and observational studies must complement each other. Quantification of cloud microphysical properties, precipitation formation process rates, and $S_o$ for different aerosol regimes using in situ data can aid modelling efforts. For example, the autoconversion and accretion rates were based on a commonly used model parameterization. The aerosol-induced changes could be compared with model output. Given the size of the dataset used, model

simulations were beyond the scope of the study. The authors agree with the reviewer that most observational studies of aerosol effects on clouds and precipitation would benefit from model support since in situ observations essentially provide snapshots of a particular cloud.

The data analysis throughout this study quantified perturbations in droplet concentration ($N_c$) and precipitation rate ($R_p$) due to droplet growth or increasing aerosol concentration ($N_a$). Figure 9 was added to link the data analysis to changes in $S_o$ as a function of $H$ under different aerosol regimes. The study addressed hypotheses about variations in $S_o$ under different above- or below-cloud $N_a$ (Duong et al., 2011; Jung et al., 2016). We do not know of a prior study that did this using in situ measurements. Observational analysis of variables/processes affecting $S_o$ for the aerosol regimes should justify the current format as a research article. Further, this study is similar in scope to many other papers that used observational data to evaluate process-based hypotheses without inclusion of modeling results.

- Are the scientific methods and assumptions valid and clearly outlined?

- Are the results sufficient to support the interpretations and conclusions?

- The paper uses specific statistical terminology (e.g. "95% Confidence Interval") which inherently implies parameters are known to exhibit normal distributions when properly sampled. Is this valid? Is this approach needed?

95% confidence intervals were used to provide statistical confidence to the comparisons between aerosol and cloud properties from different aerosol regimes. The average values for the parameters were provided throughout the study (Table 5 was updated to add the averages). Since every variable may not be normally distributed, the addition of average values allows the reader to directly compare the average values rather than using 95% confidence intervals. It is noted that the 95% confidence intervals include this direct difference between the average values regardless of the shape of the distribution.

- Adiabatic approximations of Brenguier are used. Limitations should be discussed/quantified.

The adiabatic value of liquid water content (LWC) was used as a maximum threshold to select the cloud probe to be used for analysis of data from ORACLES 2016. The impact of choosing the CAS over the PDI was discussed in detail within author responses to Reviewer Comment 1. Other discussions based on the adiabatic model did not affect data analysis or the conclusions. These discussions (Lines 244 to 252 and Lines 256 to 271) were removed for brevity.

The following sentence was added to the paper: "$LWC_{ad}$ can be used to compare LWC from different probes since it is derived using environmental conditions without any input from the cloud probes."

- Measurement uncertainties are not presented alongside observations. E.g. What is the estimated uncertainty in measurements of droplet effective radius, and how does this

relate to changes between cloud base/top, and also between the aerosol regimes studied?

The uncertainties associated with cloud probes were discussed in the supplement. Estimates of LWC from cloud droplet size distributions were validated against an independent measurement of LWC from a hot-wire. Previous work has shown a 15 to 20 % uncertainty in $N_c$ can result in up to a 50 % uncertainty in LWC (Lance, 2012). These estimates were consistent with the differences in $N_c$ and LWC between cloud probes used during ORACLES (see supplement).

A sizing uncertainty within 20 % could be expected for droplets larger than 5 μm from CAS and CDP, 50 μm from 2D-S, and 750 μm from HVPS-3 (Baumgardner et al., 2017). For differences between $R_e$ across aerosol regimes (below 2 μm) and changes in $R_e$ from cloud base to top (below 3 μm), the uncertainty can be assumed to be constant. Since relative changes in cloud properties were quantified, measurement uncertainties would have a minor impact on the results. The following sentences were added to the paper:

"Measurement uncertainties in droplet sizes were expected to be within 20 % for droplets with $D > 5$ μm from the CAS and the CDP, $D > 50$ μm from the 2D-S, and $D > 750$ μm from the HVPS-3 (Baumgardner et al., 2017)."

"The relative differences between the LWC$_{ad}$ and the LWC estimates from cloud probes provide a measure of the uncertainty associated with using one probe over the other for data analysis."

- Calculations in the paper suggest the thinnest clouds have large precipitation sensitivity to aerosols. This seems odd given these thin clouds have nominally the same droplet concentrations as thicker clouds, but only have the smaller droplets. This raises concerns with how data are handled and the overall validity of conclusions drawn from the analysis. From the text I don't fully understanding what was done here with the data to determine the precipitation susceptibility, so maybe my interpretation is wrong, but I speculate it is a result of using outputs from regression analysis which are statistically meaningless. If this is true, the paper is presenting misleading results which is very undesirable. If this is not true it needs making clearer.

The concerns are addressed by providing specific justifications for the methodology used, data interpretation, and observations of high $S_o$ for thin clouds:

Methodology: The best fit slope from a regression between $\ln(N_c)$ and - $\ln(R_p)$ was used to quantify $S_o$ following Eq. 8. $S_o$ was quantified for different populations of cloud profiles classified based on $H$ to quantify $S_o$ as a function of $H$. This is consistent with previous studies of $S_o$ using airborne measurements (e.g., Jung et al., 2016).

Data interpretation: The authors assume the reviewer is referring to low correlation coefficient ($R$) values between $\ln(N_c)$ and $\ln(R_p)$ when they say regression outputs are "statistically

meaningless". The values of $R$ between 0.3 to 0.6 were consistent with previous studies (e.g., Jung et al., 2016) and were statistically significant. Low values of $R$ were observed because $N_c$ was calculated for the entire droplet size distribution while $R_p$ was calculated for drizzle drops (diameter $D > 50$ μm). If $R_p$ was calculated for the entire size distribution, the values for $R$ increased. However, including smaller droplets within $R_p$ is not useful since the smaller droplets would have little chance of precipitating.

Thin clouds having high $S_o$: The quantification of high $S_o$ for thin clouds in the cleanest conditions and $S_o$ close to zero for thin clouds in polluted conditions is consistent with previous studies (discussed in Section 6.4). The reviewer may be referring to cloud profiles with $H < 129$ m from the Separated-low $N_a$ regime (Fig. 8b) since these profiles contributed to the high $S_o$ for thin separated profiles (Fig. 8a). Figure 7 (c, d) shows these profiles had very low $N_c$ and the highest $R_p$, on average, compared to thin clouds from other aerosol regimes. These profiles had fewer droplets and these droplets more frequently had $D > 50$ μm. It is reasonable to see high values of $S_o$ in these conditions (see Figure 9 which was added to the text).

- The stratiform clouds are shown to be around 200m thick, and often occur in the vicinity of convective clouds. Is direct comparison of high resolution in-situ datasets with relatively coarse resolution ERA5 reanalysis data sufficient to untangle effects of meteorology? What small-scale/local variations in SST could you expect based on other studies? What are the actual sizes/resolutions of ERA5 grid boxes in units relatable to the observations? Can ERA5 resolve the inversions etc? The correlations in Fig10b between LWP from in-situ and ERA5 are poor, which casts a large doubt over the validity over the in-situ LWP vs SST/LTS/EIS from ERA5. Why aren't in-situ observations of inversion strength analysed?

Reanalysis data have been used in recent studies to constrain environmental conditions and their impact on LWP and/or aerosol-cloud interactions (Douglas and L'Ecuyer 2019; 2020). The advantage of using reanalysis data was that calculation of LTS and EIS was consistent across all profiles. This was desirable given the main purpose of LTS and EIS was comparisons between aerosol regimes.

The horizontal resolution of ERA5 reanalysis was 0.25 degrees latitude and longitude (Hersbach et al., 2020) which is about 20 km. For closed cell marine stratocumulus, horizontal heterogeneity over this distance can be assumed to be low. Based on the reanalysis temperature at different pressure levels, the model was able to resolve the inversion near cloud tops for the co-located in situ profiles (Fig. 1).

Low correlation between in situ LWP and ERA5 LWP did not have a dependence on the thermodynamic parameters used to determine EIS or LTS. Ahlgrimm et al. (2009) showed biases in cloud properties from the model were due to assumptions within the model parameterization They found improved correlation between model LWP and ground-based LWP when the autoconversion-accretion parameterization was updated.

A sawtooth pattern followed for cloud sampling during ORACLES meant the aircraft frequently flew only about 100 m above or below the cloud layer (Fig. 2) Further, there were concerns with airborne measurements of thermodynamic parameters during descents into cloud (Gupta et al., 2021). This meant that thermodynamic parameters needed to calculate LTS or EIS were not available from in situ measurements near every profile.

- From the very beginning this paper places emphasis on the role of aerosols from above cloud and their ability to modify clouds via entrainment etc. There is no discussion of the potential for the boundary layer being polluted with Biomass Burning aerosols in its own right, without the requirement for entrainment from above the BL inversion. Is there data showing the transition of the BL from clean to polluted as aerosol mix downwards? If so it would be very useful to show it.

It is unlikely there were sources of biomass-burning aerosols over the southeast Atlantic Ocean. Continental aerosols reached the marine boundary layer through cloud-top entrainment or entrainment into a clear boundary layer. Evidence to support this is provided:

Biomass burning aerosols are lofted into the free troposphere over the continent (Gui et al., 2021). The aerosol layer is transported over the southeast Atlantic by mid-tropospheric winds (Adebiyi and Zuidema, 2016). Back-trajectory analysis has shown polluted above-cloud airmasses originate from high altitudes over the continent while clean below-cloud airmasses originating from the boundary layer in the southeast (Gupta et al., 2021; Miller et al., 2021). The altitude of the aerosol layer near Ascension Island also tends to increase from July to October (Zhang and Zuidema, 2021).

The following text was added to the introduction section: "Biomass-burning aerosols from southern Africa are lofted into the free troposphere (Gui et al., 2021) and transported over the southeast Atlantic by mid-tropospheric winds where the aerosols overlay an extensive MSC deck that exists off the coast of Namibia and Angola (Adebiyi and Zuidema, 2016; Devasthale and Thomas, 2011)."

The boundary layer could be polluted due to entrainment prior to in situ observations (Diamond et al., 2018). Ground-based observations from Ascension Island have shown clean boundary layers with elevated biomass burning trace gas concentrations during the burning season (Pennypacker et al., 2020). This suggests precipitation scavenging can lead to clean boundary layers in terms of $N_a$ despite the entrainment of biomass-burning aerosols into the boundary layer.

The following text was added to Section 6.1: "Ground-based observations from Ascension Island have shown clean boundary layers can have elevated biomass burning trace gas concentrations during the burning season (Pennypacker et al., 2020). This suggests boundary layers could be clean in terms of $N_a$ despite the entrainment of biomass-burning aerosols into the boundary layer due to precipitation scavenging of below-cloud aerosols."

- The paper filters data according to aerosol concentrations above cloud ("contact" vs "separate" using a 500cm$^{-3}$ threshold) and below cloud ("high" and "low" Na with a threshold of 350cm$^{-3}$). However, the cloud droplet concentrations in Fig 6 do not show evidence of enhancement due to above cloud aerosols for the "clean" BL cases. The only strong response in droplet concentration is when there are lots of aerosols also in the boundary layer. It seems impossible to disentangle the below and above cloud aerosols and therefore the role of entrainment and above cloud aerosols is ambiguous.

The reviewer is directed to Table 5 where the increase in $N_c$ for low boundary layers was quantified and compared with the corresponding increase in $N_c$ for polluted boundary layers. While the increase in $N_c$ was relatively smaller compared to polluted boundary layers, it was statistically significant. We agree with the reviewer's statement that it is difficult to disentangle the relative impact of above- and below-cloud aerosols. Thus, the study did not distinguish the impact of above-cloud versus below-cloud aerosols. Instead, the combined impact of above- and below-cloud aerosols was compared with the impact of above-cloud aerosols alone (Lines 437 to 443 in the original manuscript).

- There is no contextualisation of the results. For instance, are the calculated changes in re, or values of S$_o$ "large" or "small"? Are changes in these clouds due to the Biomass Burning aerosols having any meaningful impact? What have other studies found?

As stated in Section 6.3, $S_o$ for observational datasets should only be compared with observational datasets given the dependence of $S_o$ on data analysis techniques and aerosol analogues used in satellite studies (e.g., Sorooshian et al., 2009). Previous studies using observational data did not quantify $S_o$ under different aerosol regimes within a similar domain. Instead, $S_o$ has been quantified for different cloud types or regions. Therefore, it is difficult to contextualize the changes in $S_o$ presented in this study in terms of previous studies. The changes in $R_e$ and $N_c$ were consistent with previous studies which are referenced throughout the text.

- Is the description of experiments and calculations sufficiently complete and precise to allow their reproduction by fellow scientists (traceability of results)?
- Are mathematical formulae, symbols, abbreviations, and units correctly defined and used?
- Undefined formulae: Z$_N$

The following text was added (Line 304) to address this: "Figure 3 shows violin plots for cloud properties as a function of normalized height ($Z_N$), defined as $Z_N = Z - Z_B / Z_T - Z_B$."

- Confusing presentation of $\Gamma^{850}_{\eth \bullet ' \check{s},}$ in eqn 10

This part of the equation was removed to avoid confusion along with the equation that defined $\Gamma_m$ (Eq. 11). Instead, the reader was directed to Wood and Bretherton (2006) consistent with the approach followed by Douglas and L'Ecuyer (2021).

- Description of LCL is confusing and the equation is poorly formatted

LCL has been defined using the appropriate formatting.

- Some of the technical details of data processing are in figure captions, but should be included in the text.

Assuming this comment was directed at Figure 3, the caption is improved for clarity and the following description was added to the text: "The violin plots include box plots and illustrate the distribution of the data (Hintze and Nelson, 1998)." The details for data presented in every figure are provided during the corresponding discussion in the text.

- What are the "kernel density estimates" mentioned in caption for Figure 3? They are not mentioned anywhere else in the paper.

The figure shows violin plots where the width of the shaded area represents the proportion of data there (Hintze and Nelson, 1998). The description was added to the text: "The violin plots include box plots and illustrate the distribution of the data (Hintze and Nelson, 1998)."

- Is the overall presentation well structured and clear?
- Is the language fluent and precise?
- Should any parts of the paper (text, formulae, figures, tables) be clarified, reduced, combined, or eliminated?
- The paper was difficult to follow. I feel the paper is too long and lacked coherence. It is very ambitious, and the authors have covered lots of areas which are all important and related, but the balance is not quite right. The paper has lots of useful data but in its current form does not provide concrete outputs which can be used by the broader community.

The concerns with balance/coherence, presentation, and paper length were addressed:

The title of the paper was edited to emphasize the role of factors affecting $S_o$. Figure 9 was added to relate the preceding data analysis with the discussion of $S_o$ in Sections 6.3 and 6.4. The impact of perturbations in $N_c$ and $R_p$ on $S_o$ (due to droplet growth processes with $H$ or increasing aerosols) was further illustrated in a mathematical framework. Recommendations for future work were added. The following text was removed to reduce the paper length:

- Line 214 to 218: Comment on relationship between cloud top height and liquid water path adjustments associated with aerosol-cloud interactions.
- Line 244 to 252, 256 to 271: The discussion of parameters associated with adiabatic cloud optical thickness.
- Line 315, 318 to 320: Comment on aerosol influence on cloud water.
- Line 576 to 579: Definition of $\Gamma_m$.

- Most figures should be improved and are poorly rendered, and some do not have proper legends etc (e.g. Fig 10b has a 1:1 line listed as "x=y", wrong coloured text in legends).

Every figure was updated and rendered following journal guidelines (300 dpi resolution). The legends were updated: text color corrected for Fig. 3, 4, and 9 and "x=y" replaced by "1:1 line" for Fig. 10b.

- Do the authors give proper credit to related work and clearly indicate their own new/original contribution?

- Are the number and quality of references appropriate?

- Yes there are a good number of quality references. Some references are missing (e.g. description of instruments in section 2) but nothing major.

The appropriate references were added to Section 2.

- Is the amount and quality of supplementary material appropriate?

- Yes, supplementary material is of good quality and is a useful addition.

FIGURES:

[Figure]

Figure 1: Box plots of temperature from the ERA5 reanalysis at model pressure levels. The data correspond to grid boxes co-located with an in situ cloud profile used in the study.

[Figure]

Figure 2: P-3 aircraft altitude as a function of time during sawtooth flight patterns. Data are colored by accumulation mode aerosol concentration (taken from Gupta et al., 2021).

REFERENCES:

Ahlgrimm M, Randall DA, Kohler M.. Evaluating cloud frequency of ¨ occurrence and cloud-top height using spaceborne lidar observations. Mon. Weather Rev. 137: 4225–4237, 2009.

Baumgardner, D., Abel, S. J., Axisa, D., Cotton, R., Crosier, J., Field, P., Gurganus, C., Heymsfield, A., Korolev, A., Kraemer, M., Lawson, P., McFarquhar, G., Ulanowski, Z., and Um, J.: Cloud ice properties: in situ measurement challenges, Meteor. Monographs, 58, 9.1–9.23, https://doi.org/10.1175/AMSMONOGRAPHS-D-16-0011.1, 2017.

Diamond, M. S., Dobracki, A., Freitag, S., Small Griswold, J. D., Heikkila, A., Howell, S. G., Kacarab, M. E., Podolske, J. R., Saide, P. E., and Wood, R.: Time-dependent entrainment of smoke presents an observational challenge for assessing aerosol–cloud interactions over the southeast Atlantic Ocean, Atmos. Chem. Phys., 18, 14623–14636, https://doi.org/10.5194/acp-18- 14623-2018, 2018.

Douglas, A. and L'Ecuyer, T.: Quantifying variations in shortwave aerosol–cloud–radiation interactions using local meteorology and cloud state constraints, Atmos. Chem. Phys., 19, 6251–6268, https://doi.org/10.5194/acp-19-6251-2019, 2019.

Douglas, A. and L'Ecuyer, T.: Quantifying cloud adjustments and the radiative forcing due to aerosol–cloud interactions in satellite observations of warm marine clouds, Atmos. Chem. Phys., 20, 6225–6241, https://doi.org/10.5194/acp-20-6225-2020, 2020.

Douglas, A. and L'Ecuyer, T.: Global evidence of aerosol-induced invigoration in marine cumulus clouds, Atmos. Chem. Phys., 21, 15103–15114, https://doi.org/10.5194/acp-21-15103-2021, 2021.

Duong, H. T., Sorooshian, A., and Feingold, G.: Investigating potential biases in observed and modeled metrics of aerosol-cloud-precipitation interactions, Atmos. Chem. Phys., 11, 4027–4037, https://doi.org/10.5194/acp-11-4027-2011, 2011.

Gui, K., Che, H., Zheng, Y., Zhao, H., Yao, W., Li, L., Zhang, L., Wang, H., Wang, Y., and Zhang, X.: Three-dimensional climatology, trends, and meteorological drivers of global and regional tropospheric type-dependent aerosols: insights from 13 years (2007–2019) of CALIOP observations, Atmos. Chem. Phys., 21, 15309–15336, https://doi.org/10.5194/acp-21-15309-2021, 2021.

Gupta, S., McFarquhar, G. M., O'Brien, J. R., Delene, D. J., Poellot, M. R., Dobracki, A., Podolske, J. R., Redemann, J., LeBlanc, S. E., Segal-Rozenhaimer, M., and Pistone, K.: Impact of the variability in vertical separation between biomass burning aerosols and marine stratocumulus on cloud microphysical properties over the Southeast Atlantic, Atmos. Chem. Phys., 21, 4615–4635, https://doi.org/10.5194/acp-21-4615-2021, 2021.

Hersbach, H., Bell, B., Berrisford, P., Hirahara, S., Horányi, A., Muñoz-Sabater, J., Nicolas, J., Peubey, C., Radu, R., Schepers, D., Simmons, A., Soci, C., Abdalla, S., Abellan, X., Balsamo, G., Bechtold, P., Biavati, G., Bidlot, J., Bonavita, M., De Chiara, G., Dahlgren, P., Dee, D., Diamantakis, M., Dragani, R., Flemming, J., Forbes, R., Fuentes, M., Geer, A., Haimberger, L., Healy, S., Hogan, R. J., Hólm, E., Janisková, M., Keeley, S., Laloyaux, P., Lopez, P., Lupu, C., Radnoti, G., de Rosnay, P., Rozum, I., Vamborg, F., Villaume, S., and Thépaut, J.-N.: The ERA5 Global Reanalysis, Q. J. Roy. Meteor. Soc., 146, 730, 1999– 2049, https://doi.org/10.1002/qj.3803, 2020.

Hintze, J. L. and Nelson, R. D.: Violin Plots: A Box Plot-Density Trace Synergism, Am. Stat., 52, 181–184, 1998.

Jung, E., Albrecht, B. A., Sorooshian, A., Zuidema, P., and Jonsson, H. H.: Precipitation susceptibility in marine stratocumulus and shallow cumulus from airborne measurements, Atmos. Chem. Phys., 16, 11395–11413, https://doi.org/10.5194/acp-16-11395-2016, 2016.

Lance, S.: Coincidence Errors in a Cloud Droplet Probe (CDP) and a Cloud and Aerosol Spectrometer (CAS), and the Improved Performance of a Modified CDP, J. Atmos. Ocean. Tech., 29, 1532– 1541, doi:10.1175/JTECH-D-11-00208.1, 2012.

Miller, R. M., McFarquhar, G. M., Rauber, R. M., O'Brien, J. R., Gupta, S., Segal-Rozenhaimer, M., Dobracki, A. N., Sedlacek, A. J., Burton, S. P., Howell, S. G., Freitag, S., and Dang, C.: Observations of supermicron-sized aerosols originating from biomass burning in southern Central Africa, Atmos. Chem. Phys., 21, 14815–14831, https://doi.org/10.5194/acp-21-14815-2021, 2021.

Pennypacker, S., Diamond, M., and Wood, R.: Ultra-clean and smoky marine boundary layers frequently occur in the same season over the southeast Atlantic, Atmos. Chem. Phys., 20, 2341–2351, https://doi.org/10.5194/acp-20-2341-2020, 2020.

Sorooshian, A., Feingold, G., Lebsock, M. D., Jiang, H., and Stephens, G. L.: On the precipitation susceptibility of clouds to aerosol perturbations, Geophys. Res. Lett., 36, L13803, doi:10.1029/2009GL038993, 2009.

Wood, R. and Bretherton, C. S.: On the relationship between stratiform low cloud cover and lower-tropospheric stability, J. Climate, 19, 6425–6432, 2006.

Zhang, J. and Zuidema, P.: Sunlight-absorbing aerosol amplifies the seasonal cycle in low-cloud fraction over the southeast Atlantic, Atmos. Chem. Phys., 21, 11179–11199, https://doi.org/10.5194/acp-21-11179-2021, 2021.

---

## Author Response (AR2)

This document includes the reviewer comments (red) and author responses (black). Author responses to anonymous referee #1 are followed by author responses to anonymous referee #3.

**Author responses to Anonymous Referee #1:**

The authors response adequately addresses my comments on the previous version of the document. However, I have a few minor comments on the revised manuscript.

1. Although the authors addressed my main concern in their response about the use of the CAS data from the 2016 campaign, I would have liked to see them briefly summarize this in the revision. Perhaps around line 217.

A brief summary was added as an appendix at the end of the paper. The following sentence was added at the end of section 6.3: "CAS data were used to represent measurements of droplets with $D < 50$ μm collected during ORACLES 2016 in the absence of CDP data. The sensitivity of $S_o$ to the use of CAS data was examined in Appendix D." The following text was added at the end of the manuscript (along with the addition of Fig. D1 and Table D1):

"Appendix D – Sensitivity of $S_o$ to the use of CAS data from ORACLES 2016

Given the differences between the CAS and PDI $N_c$ and LWC for droplets with $D < 50$ μm during ORACLES 2016 (see supplement), sensitivity tests were performed by first excluding ORACLES 2016 data and second by using PDI data to represent ORACLES 2016 size distributions for $D < 50$ μm in the $S_o$ calculations. These sensitivity tests resulted in minor changes in the trends of $S_o$ versus $H$ (Fig. D1) along with average changes in the magnitude of $S_o$ up to 0.05 (Table D1). The noted changes suggest that the discussion of trends in $S_o$ described in this study is robust as it relates to the inclusion of ORACLES 2016 data and the use of CAS data for the deployment. Since the 2016 deployment contributed about a third of the ORACLES measurements, data from the 2016 deployment were included in the study so as not to reduce the size of the dataset.

The slight decrease in $S_o$ for thick clouds ($H > 256$ m) upon removal of ORACLES 2016 data is associated with a decrease in the number of thick clouds (Table D1). The use of PDI data resulted in minor changes because $S_o$ primarily depends on $N_c$ and $R_p$. The CAS and PDI datasets had small differences in the average $N_c$ (95 % confidence intervals of 9 to 12 cm$^{-3}$) and $R_p$ was calculated using droplets with $D > 50$ μm which do not include contributions from either the CAS or the PDI. The documentation of differences between the ORACLES cloud probes (see supplement) highlights the measurement uncertainties associated with the cloud probe datasets."

2. Line 36: Suggest changing "the existing relationships between Rp and Nc in model parameterizations must be adjusted to account..." to something like "adjustments to existing relationships between Rp and Nc in model parameterizations should be considered to account..."

The sentence was updated to "These results suggest the changes in cloud microphysical properties were driven by ACIs rather than meteorological effects, and adjustments to existing

relationships between $R_p$ and $N_c$ in model parameterizations should be considered to account for the role of ACIs."

3. Line 515: Should this be Fig 7 instead of Fig 8?

This was changed.

4. Line 592: I think, but am not entirely convinced, that I understand what the authors are trying to convey here with reference to the new Fig 9. Suggest they expand the description in the text and figure caption a little. So that it is clear how the figure shows "the impact of deltaNc or deltaRp on S0 depends on the original...."

The following text was added at the end of the paragraph: "Figure 9 shows the impact of $\Delta N_c$ and $\Delta R_p$ on $S_o$ depends on the original values for $N_c$ and $R_p$ as the same $\Delta N_c$ or $\Delta R_p$ can have an opposing effect on $S_o$. For example, a decrease in $N_c$ at point 1 would decrease the slope and the $S_o$ value while the same decrease in $N_c$ at point 2 would increase the slope and the $S_o$ value."

**Author responses to Anonymous Referee #3:**

The authors have done a good job to address reviewer concerns. I had some minor suggestions:

Line ~95-96: when you write "unique meteorological conditions", what is meant be this? be specific rather than citing other papers.

The papers were cited to point the reader to information on field campaigns mentioned at the start of the sentence. The paragraph text was rearranged, and the unique conditions are introduced in the next two sentences:

"Recent field campaigns focused on studying ACIs over the southeast Atlantic Ocean because unique meteorological conditions are present in the region (Zuidema et al., 2016; Redemann et al., 2021). Biomass-burning aerosols from southern Africa are lofted into the free troposphere (Gui et al., 2021) and transported over the southeast Atlantic by mid-tropospheric winds where the aerosols overlay an extensive MSC deck that exists off the coast of Namibia and Angola (Adebiyi and Zuidema, 2016; Devasthale and Thomas, 2011). The above-cloud aerosol plume is associated with elevated water vapor content (Pistone at al., 2021) which influences cloud-top humidity and dynamics following the mechanisms discussed by Ackerman et al. (2004)."

The title seems to address only a fraction of the paper. Another title would be more helpful that fully captures everything that is presented.

The title of the paper was changed to: "Factors Affecting Precipitation Formation and Precipitation Susceptibility of Marine Stratocumulus with Variable Above and Below-Cloud Aerosol Concentrations over the Southeast Atlantic"

It seems there are sometimes negative precip susceptibility values. This is interesting and can be discussed briefly in light of others who have shown this too and attributed it to possible influence of giant particles: https://doi.org/10.1002/2016JD026019

The following text was added in Section 6.4:

"An airborne investigation of marine stratocumulus off the Californian coast attributed negative values of $S_o$ to the influence of giant cloud condensation nuclei (Dadashazar et al., 2017). The authors hypothesized the low statistical significance of the negative estimate of $S_o$ could be associated with precipitation suppression by aerosol particles."

Regarding these statements: "The ORACLES dataset addresses the "lack of long-term data sets needed to provide statistical significance for a sufficiently large range of aerosol variability influencing specific cloud regimes over a range of macrophysical conditions" (Sorooshian et al., 2010)."...My suggest is to tone this down a bit since this one dataset doesn't fully address this issue. It is a step in the right direction, but still there is room for improvement with more flights and to gather more statistics.

This sentence was moved to the end of the paragraph and changed to:

"The ORACLES dataset can be combined with future investigations of marine stratocumulus to address the "lack of long-term data sets needed to provide statistical significance for a sufficiently large range of aerosol variability influencing specific cloud regimes over a range of macrophysical conditions" (Sorooshian et al., 2010)."

OTHER CHANGES:

1. The first affiliation was updated to reflect a name change for the organization.
2. Table formatting was changed to remove colors following journal recommendation.
3. Line 198: O'Brien et al. (2021, in prep) changed to O'Brien et al. (2022, in prep)
4. The following reference was added:
   "Dadashazar, H., Wang, Z., Crosbie, E., Brunke, M., Zeng, X., Jonsson, H., Woods, R. K., Flagan, R. C., Seinfeld, J. H., and Sorooshian, A.: Relationships between giant sea salt particles and clouds inferred from aircraft physicochemical data, J. Geophys. Res.-Atmos., 122, 3421–3434, https://doi.org/10.1002/2016JD026019, 2017."